# Zero-Mean Regularized Spectral Contrastive Learning: Implicitly Mitigating Wrong Connections in Positive-Pair Graphs

**Xiong Zhou, Xianming Liu,**\* **Feilong Zhang, Gang Wu, Deming Zhai & Junjun Jiang**
Faculty of Computing, Harbin Institute of Technology
`{cszx,csxm,flzhang,gwu,zhaideming,jiangjunjun}@hit.edu.cn`

**Xiangyang Ji**
Department of Automation, Tsinghua University
`xyji@tsinghua.edu.cn`

## Abstract

Contrastive learning has emerged as a popular paradigm of self-supervised learning that learns representations by encouraging representations of positive pairs to be similar while representations of negative pairs to be far apart. The spectral contrastive loss, in synergy with the notion of positive-pair graphs, offers valuable theoretical insights into the empirical successes of contrastive learning. In this paper, we propose incorporating an additive factor into the term of spectral contrastive loss involving negative pairs. This simple modification can be equivalently viewed as introducing a regularization term that enforces the mean of representations to be zero, which thus is referred to as *zero-mean regularization*. It intuitively relaxes the orthogonality of representations between negative pairs and implicitly alleviates the adverse effect of wrong connections in the positive-pair graph, leading to better performance and robustness. To clarify this, we thoroughly investigate the role of zero-mean regularized spectral contrastive loss in both unsupervised and supervised scenarios with respect to theoretical analysis and quantitative evaluation. These results highlight the potential of zero-mean regularized spectral contrastive learning to be a promising approach in various tasks.

## 1 Introduction

Contrastive learning has emerged as one of the most prominent self-supervised learning paradigms, which offers promising representations that can be adapted to diverse downstream tasks [5, 6, 7, 8, 9, 19, 23, 30, 55, 56]. Contrastive losses serve as the training objectives for contrastive learning, encouraging the learning of representations invariant to data augmentations by maximizing the similarity between features from different distortions of the same images. In addition to unsupervised training, contrastive losses are extended to the fully-supervised setting as an alternative to the classical cross-entropy loss to effectively leverage label information [26]. This extension enables contrastive learning to achieve state-of-the-art performance on various supervised learning tasks.

Researchers have attempted to offer theoretical understanding of the empirical successes of contrastive learning [1, 46, 50, 51, 53]. Some works provide mathematical analysis, which, however, are under an impractical assumption that two views are somewhat independent conditioned on the label [29, 43, 52]. Instead, spectral contrastive loss (SpeCL) presents solid theoretical foundations without requiring conditional independence but on a more realistic data property that there is continuity of the population data within the same class [19, 20]. The core concept in SpeCL is the positive-pair graph on data, where nodes are augmented samples and the edge between two nodes is weighted as the probability of encountering them as a positive pair. By applying spectral decomposition on the adjacency matrix defined on the population augmentation graph, SpeCL builds the relationship between the eigenvector matrix and the learned representations. SpeCL naturally exhibits *sample-*

---

\*Correspondence to: Xianming Liu <csxm@hit.edu.cn>

*contrastive* property, while the covariance regularization term within the equivalent form [20] can be seen as contrastive between the dimensions of the representations, and thus coincides with the *dimension-contrastive* property. Therefore, SpeCL serves as a bridge between sample-contrastive and dimension-contrastive properties, offering a pathway to unify them. Moreover, SpeCL resembles state-of-the-art contrastive losses, such as Barlow Twins loss [57] and VICReg loss [1, 2].

Although SpeCL has appealing theoretical advantages, there are still two limitations carefully considered in this paper. Firstly, SpeCL intuitively requires the orthogonality of representations between negative pairs. However, relaxing the orthogonality could potentially enhance the discriminativeness of representations. Secondly, the pairwise similarities among the representations learned by SpeCL are determined by the connections within the positive-pair graph. Unfortunately, the presence of false positive pairs, which may be due to noisy views in self-supervised learning [10] or noisy labels in supervised learning [63], can lead to wrong connections (i.e., unfavorable edge weights). Identifying and correcting these wrong connections is scarcely possible since the underlying structure of the graph is unknown. To overcome these limitations, we propose to incorporate an additional factor $\tau$ into the term of SpeCL involving negative pairs, which is simple yet effective to relax the orthogonality and coincides with the motivation of margin-based losses [12, 33, 34, 54, 61]. By algebraic manipulation, it can be found that our scheme is equivalent to regularizing the mean of representations to be zero, thus referred to as *zero-mean regularization*. Furthermore, we would establish that the introduction of $\tau$ explicitly performs a uniform reduction in all positive-pair weights, which modifies pairwise similarities between positive pairs and implicitly mitigates the adverse effects of wrong connections.

To demonstrate the effectiveness of zero-mean regularization in mitigating adverse effects of wrong connections, we provide theoretical investigations in both unsupervised and supervised scenarios: unsupervised domain adaptation (UDA) and supervised learning with noisy labels. For contrastive pretraining based UDA, wrong connections occurs when positive pairs are built from different domains and classes, we prove that the incorporation of $\tau$ tightens the downstream error on target domains in the form of multiplying a factor $(1-\tau)^2$. For supervised learning with noisy labels, false positive pairs originate from the misguidance of label noise. We first establish the supervised version of SpeCL and show that the global minimizer exhibits a geometric structure similar to the recently discovered Neural Collapse [39], and we prove that zero-mean regularization can mitigate label noise by implicitly reducing mislabeled weights in the noise transition matrix.

The main contributions of our paper can be summarized as follows:

- We propose to incorporate an additional factor into spectral contrastive loss involving negative pairs, which is equivalent to the zero-mean regularization term. We also show that zero-mean regularization implicitly performs a uniform weight reduction in positive pairs.

- We investigate the role of zero-mean regularized spectral contrastive loss on spectral contrastive pretraining-based unsupervised domain adaptation and offer theoretical proof that zero-mean regularization can tighten the error bound by multiplying a factor less than one.

- We establish the supervised version of spectral contrastive loss and derive the closed-form optimal representations, which resembles the Neural Collapse phenomenon and suggests using class-mean features as classifier. We further prove that zero-mean regularization can mitigate label noise by implicitly reducing mislabeled weights in the noise transition matrix.

## 2 PRELIMINARY

We consider datasets containing samples drawn from the input space $\mathcal{X}$ with the distribution $p_{\text{data}}$ that can be partitioned into $r$ different classes with the label function $y(\cdot) : \mathcal{X} \to [r]$. The representation $f(x) \in \mathbb{R}^d$ is extracted by a mapping $f : \mathcal{X} \to \mathbb{R}^d$ that is usually characterized by a number of parameterized compositions, where $d$ is the representation dimensionality.

**Spectral Contrastive Loss.** The spectral contrastive loss is formulated as [19]:

$$\mathcal{L}(f) = -2 \cdot \mathbb{E}_{(x,x^+) \sim p_{\text{pos}}} \left[ f(x)^\top f(x^+) \right] + \mathbb{E}_{(x,x^-) \sim p_{\text{data}}} \left[ \left( f(x)^\top f(x^-) \right)^2 \right]$$

$$= \underbrace{\mathbb{E}_{x,x^+} \left[ \left\| f(x) - f(x^+) \right\|_2^2 \right]}_{\mathcal{R}_0(f)} + \underbrace{\left\| \mathbb{E}_x \left[ f(x) f(x)^\top \right] - \mathrm{I} \right\|_F^2}_{\mathcal{R}_1(f)} + const, \tag{2.1}$$

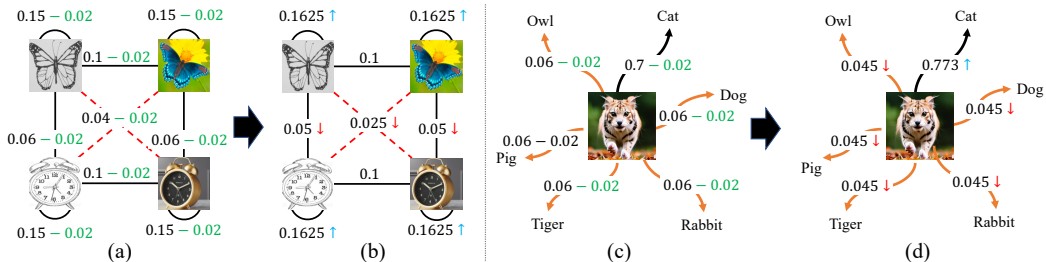

Figure 1: Illustration of the uniform reduction of positive-pair graphs implied in self-supervised learning (a-b) and learning with noisy labels (c-d). (a) Edge weights denote connectivity (probability of sampling the endpoints as a positive pair). The red dashed edges means adverse connections since the similarity of two very different samples will hinder the discriminativeness of representations. When these weights in the positive-pair graph are subtracted with $0.02$ and then normalized, resulting in (b) The red dashed edges are alleviated as the weights decrease, while the consistency is enhanced as self-loop weights increases. (c) In the presence of label noise, positive-pair weights for supervised spectral contrastive learning can be viewed as the label noise transition matrix. Here, edges weights denote the probabilities of annotating the sample to every labels. Orange arrows indicate the adverse connections which introduce label noise. By subtracting these weights by $0.02$ and normalizing them, we obtain (d) The orange connections are mitigated as their weights are reduced.

where $(x, x^+) \sim p_{\text{pos}}$ is a pair of augmentations of the same data, $(x, x^-)$ is a pair of independently random augmented data, and I is the identity matrix. The second equality, as identified in [20], reveals that the spectral contrastive loss has two terms: (i) $\mathcal{R}_0(f)$, namely the invariance term, measures the $\ell_2$ distance between positive pairs, which can also be expressed as the Dirichlet energy of representations on the positive-pair graph [1]; (ii) $\mathcal{R}_1(f)$, restricts the representation covariance towards the identity matrix, similar to Barlow Twins loss [57] and VICReg loss [1, 2]. Spectral contrastive loss naturally falls under the category of a *sample-contrastive* method, which also exhibits the *dimension-contrastive* characteristic due to the existence of the $\mathcal{R}_1(f)$ term.

**Limitations.** Although spectral contrastive loss achieves promising theoretical and quantitative results, there are still some limitations needed to be carefully considered: (*i*) SpeCL intuitively requires the orthogonality of representation between negative pairs, while the InfoNCE loss [38] used in SimCLR [6] and MoCo [23] aims to make them opposite. Relaxing the orthogonality constraint would bring SpeCL closer to these prior works and may also enhance the discriminativeness of representations, particularly for supervised contrastive learning in Section 3.3. (*ii*) The pairwise similarities among the representations acquired through SpeCL are dictated by the relationships within the positive-pair graph. However, rectifying incorrect connections proves to be a formidable task, given that the inherent structure of the graph remains elusive. SpeCL does not incorporate any mechanism to mitigate the detrimental impact of connections stemming from erroneous positive pairs, which frequently arise in various tasks due to the inclusion of noisy views [10], noisy labels [63], and other contributing factors, as illustrated in Figure 1.

## 3 ZERO-MEAN REGULARIZED SPECTRAL CONTRASTIVE LEARNING

In this section, we introduce in detail our main contribution—zero-mean regularization for SpeCL. We explain its motivation and how it can be applied in SpeCL for alleviating detrimental connections in positive-pair graph on unsupervised domain adaptation and learning with noisy labels. We also provide more clarification in Appendix A.

### 3.1 ZERO-MEAN REGULARIZATION

We introduce the *zero-mean regularization* on SpeCL [19], which is simply incorporating an additional factor $\tau$ in the negative part of the original SpeCL. The population objective is formulated as:

$$\mathcal{L}(f; \tau) = -2 \cdot \mathbb{E}_{(x, x^+) \sim p_{\text{pos}}} \left[ f(x)^\top f(x^+) \right] + \mathbb{E}_{(x, x^-) \sim p_{\text{data}}} \left[ \left( f(x)^\top f(x^-) + \tau \right)^2 \right] \quad (3.1)$$

It can be found that, compared with Equation 2.1, the only modification is the introduction of the additive factor $\tau$. To more clearly show the role of $\tau$, we further derive its equivalent form:

$$\mathcal{L}(f;\tau) = \underbrace{\mathbb{E}_{x,x^+}\left[\left\|f(x) - f(x^+)\right\|_2^2\right]}_{\mathcal{R}_0(f)} + \underbrace{\left\|\mathbb{E}_x\left[f(x)f(x)^\top\right] - \mathrm{I}\right\|_F^2}_{\mathcal{R}_1(f)} + 2\tau \cdot \underbrace{\left\|\mathbb{E}_x\left[f(x)\right]\right\|_2^2}_{\mathcal{R}_2(f)} + const,$$

(3.2)

where $\mathcal{R}_2(f)$ regularizes the mean of representations to be zero and thus is referred to as *zero-mean regularization*, and $\tau \geq 0$ is actually the trade-off parameter that controls the regularization strength.

**Remark.** Zero-mean regularization can effectively alleviate the limitations of SpeCL: (*i*) The introduction of $\tau$ is initially motivated by the observation that SpeCL requires orthogonality of representations between negative pairs, which, in the supervised scenario (as described in Section 3.3), is roughly requiring the orthogonality between different classes. However, prior work has demonstrated that the representations learned by cross entropy and mean-squared error during the terminal phase of training usually exhibit the Neural Collapse phenomenon [17, 36, 39, 61, 65], which shows that the representations of different classes are maximally distant and form a simplex Equiangular Tight Frame (ETF)[1]. Intuitively, the introduction of $\tau$ can encourage the representations of negative pairs to have larger angles and thus makes them more discriminative. (*ii*) Spectral contrastive learning is equivalent to spectral decomposition on a positive-pair graph. The introduction of $\tau$ implicitly reduces all these positive-pair weights uniformly in Equation 3.3, which does not change the magnitude relationship between positive-pair weights but alters the ratio between them, as illustrated in Figure 1. This explicitly modifies pairwise similarities between representations, which further indicates the benefits in decoupling the class and domain information for contrastive pretraining based unsupervised domain adaptation (cf. Section 3.2) and mitigating label noise by reducing mislabeled weights in the noise transition matrix (cf. Section 3.3.2).

In the following, we introduce in detail zero-mean regularization works in both unsupervised and supervised versions of SpeCL. We carefully discuss in theory how $\tau$ is helpful in downstream unsupervised domain adaptation and supervised classification with noisy labels.

## 3.2 UNSUPERVISED DOMAIN ADAPTATION

In the UDA setting, we have access to labeled data from a source domain and unlabeled data from target domains, and the goal is to achieve good performance on target domains [4, 41, 42]. We consider a multi-way classification problem, where $r$ is the number of classes, $p$ is the number of domains, and $n$ is the number of examples in each class of each domain. The total number of data is $N = rpn$. For data $x \in \mathcal{X}$, we use $d_x \in [p]$ and $y_x \in [r]$ to denote its domain and class, respectively.

**Contrastive Pre-training based UDA.** While conventional UDA methods typically leverage the intuition of learning domain-invariant representations [14, 45, 49, 58], recent research by Shen et al. [44] has demonstrated that a model first pre-trained on both the source and target domains with SpeCL and then fine-tuned using labeled source data, can yield comparable or superior results to strong UDA methods. Furthermore, the authors have theoretically proven that spectral contrastive pre-training enables the learning of representations that vary substantially across domains, while still generalizing to the target domain by disentangling domain and class information, instead of merely satisfying domain invariance. Following the setup of [44], we also consider a stochastic block model [24] for the positive-pair graph, where the probability of existence of an edge between $x$ and $x'$ is: (1) $\rho$ if $d_x = d_{x'}$ and $y_x = y_{x'}$, (2) $\alpha$ if $d_x \neq d_{x'}$ and $y_x = y_{x'}$, (3) $\beta$ if $d_x = d_{x'}$ and $y_x \neq y_{x'}$, (4) $\gamma$ if $d_x \neq d_{x'}$ and $y_x \neq y_{x'}$. Let $A \in \mathbb{R}^{N \times N}$ be the adjacency matrix of the graph. Let $d = p + r - 1$ be the feature dimension [2] and $f : \mathcal{X} \to \mathbb{R}^d$ be the representation model.

---

[1] An ETF is a collection of vectors $V = [v_1, v_2, ..., v_r] \in \mathbb{R}^{d \times r}$ having equal lengths and equal, maximally separated pair-wise angles. In the setting of $r \leq d + 1$, we have $V^\top V = C(\mathrm{I} - \frac{1}{r}\mathbf{1}\mathbf{1}^\top)$ [39, 48, 62].

[2] The reason we assume $d = p + r - 1$ is to facilitate a better derivation and arrive at more relevant conclusions. As depicted in the proof of Theorem 4.1, there are exactly $p + r - 1$ nonzero eigenvalues in $\tilde{A}$. By assuming $d = p + r - 1$, we can easily write the $d$-rank approximation of $\tilde{A}$ according to spectral decomposition. This also reveals that the effective dimension of the feature space is $p + r - 1$. In the scenario where $d > p + r - 1$, we can directly set the values of excess dimensions to zero without compromising the validity of our conclusion. However, we acknowledge that this assumption may appear restrictive. In cases where $d < p + r - 1$, the corresponding derivation becomes intractable, making it challenging to obtain the exact form of $\tilde{A}_d$.

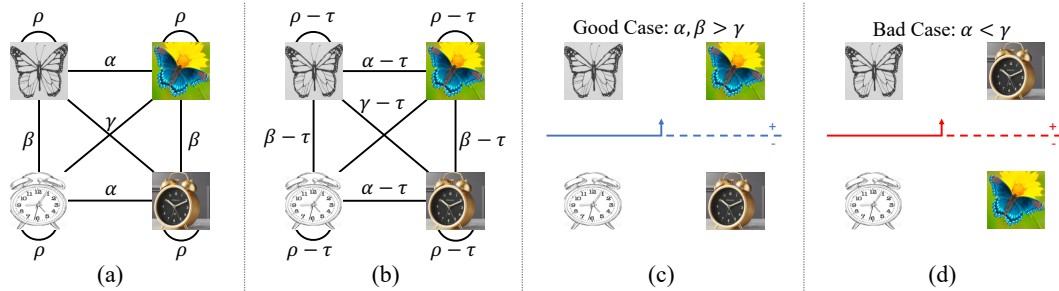

Figure 2: The illustration primarily originates from [44]. (a) Illustrative example in binary classification with two domains, where each class-domain pair is a node in the positive-pair graph. Edge weights denote the probability of sampling the endpoints as a positive pair. $\rho$ denotes the probability of sampling a pair of the same domain and class. (b) The edge weights are subtracted by a constant $\tau < \gamma$, which does not change the magnitude relationship of the edge weights but simply changes the ratio between them. (c) When $\alpha$ and $\beta$ are greater than $\gamma$, the features are oriented so that a source-trained linear classifier generalizes to the target domain. The class and domain information are disentangled along the vertical and horizontal axes, respectively. (d) When $\alpha < \gamma$, the target features are flipped and the source-trained classifier can not generalize. As can be seen, the weights of pairs belonging to the same domain and class is relatively increased, while the weights of different domains or classes are relatively diminished, *i.e.*, $\frac{\rho-\tau}{\max\{\alpha,\beta\}-\tau} > \frac{\rho}{\max\{\alpha,\beta\}}$ when $\rho > \max\{\alpha,\beta\}$, which benefits in terms of decoupling the class and domain information. Furthermore, the connections of pairs from different domains and different classes in the graph are also weakened, *i.e.*, $\frac{\min\{\alpha,\beta\}-\tau}{\gamma-\tau} > \frac{\min\{\alpha,\beta\}}{\gamma}$, which helps to improve discriminativeness across domains and classes.

**The Uniform Weight Reduction in Positive-Pair Graphs.** Let $|E|$ denote the total number of edges, according to the definition of SpeCL in Equation 3.1, we can rewrite the loss function as

$$-2\sum_{x,x'}\frac{A_{xx'}}{|E|}f(x)^\top f(x') + \sum_{x,x'}\frac{1}{N^2}\left(f(x)^\top f(x') + \tau\right)^2$$

$$= \left\|\frac{N}{|E|}\cdot\left(A - \frac{\tau|E|}{N^2}11^\top\right) - \left(\frac{1}{\sqrt{N}}\cdot F\right)\left(\frac{1}{\sqrt{N}}\cdot F\right)^\top\right\|_F^2 + const, \quad (3.3)$$

where $F \in \mathbb{R}^{N\times k}$ is the matrix whose the $x$-th row contains $f(x)^\top$. We note that the introduction of $\tau$ reduces the positive-pair weights uniformly, *i.e.*, all the entries in $A$ are subtracted by $\frac{\tau|E|}{N^2}$. As illustrated in Figure 2, subtracting a suitable positive constant from the edge weights preserves their magnitude relationship while altering the ratio between them. Specifically, the weights of pairs belonging to the same domain and class are relatively increased, whereas the weights of pairs from different domains or classes are relatively decreased. This adjustment explicitly modifies pairwise similarities between representations, leading to further benefits in terms of decoupling the class and domain information. Additionally, the connections of pairs from different domains and different classes in the graph are weakened, which enhances discriminativeness across domains and classes. We further provide theoretical analysis that $\tau$ can tighten the downstream error on the target domains.

In UDA, we care about the error on target domains. Let $\mathcal{S} = \{x \in \mathcal{X} : d_x = 1\}$ and $\mathcal{T} = \{x \in \mathcal{X} : d_x \neq 1\}$ be the source and target domains, respectively. Given the labeled source domain data, we learn the linear classifier with a pre-trained representation model $f$:

$$\hat{b} = \arg\min_{b\in\mathbb{R}^{d\times r}} \sum_{x\in\mathcal{S}}\left(\|b^\top f(x) - \vec{y}_x\|_2^2 + \eta\|b\|_F^2\right), \quad (3.4)$$

where $\vec{y}_x = e_{y_x} - \frac{1}{r}\cdot\mathbf{1} \in \mathbb{R}^r$ is the mean-zero one-hot embedding of the label, $\eta > 0$ is the regularization strength. For data $x \in \mathcal{T}$, we use $\text{pred}(x) = \arg\max_i(\hat{b}^\top\hat{f}(x))_i$ as the predictor, where $\hat{f}$ achieves the global minimum of Equation 3.3. Let $\text{pred} \in \mathbb{R}^{N\times r}$ be the matrix with $\hat{b}^\top f(x)$ on its $x$-th row and $\text{pred}_\mathcal{T}$ be the matrix by restricting $\text{pred}$ to the target domain.

Compared to Shen et al. [44], we provide a more general form of the error bound when training SpeCL with zero-mean regularization across multiple domains as follows:

**Theorem 3.1.** *Let $\zeta > 0$ and $\epsilon \in (0, \frac{1}{2})$ be arbitrary constants. In the above stochastic block model, assume $\rho > \max\{\alpha, \beta\}$, $\gamma < \min\{\alpha, \beta\}$, and $\tau < (\tilde{\lambda}_1(0) - \tilde{\lambda}_d(0))/\tilde{\lambda}_1(0)$. Then, there exists $\tilde{\xi} \in [1 - \epsilon, 1]$, such that for any $n \geq \Omega\left(\frac{rp}{\min\{\alpha - \gamma, \beta - \gamma\}^2}\right)$ and regularization strength $\eta \in \left(0, \frac{(\alpha - \gamma)\epsilon}{2r\rho}\right]$, with a high probability $1 - n^{-\zeta}$, we have*

$$\left\| \text{pred}_{\mathcal{T}} - \widetilde{\text{pred}}_{\mathcal{T}} \right\|_F \leq O\left(\frac{\tilde{\lambda}_1(\tau)}{\eta^2(\tilde{\lambda}_d(\tau) - \tilde{\lambda}_{d+1}(\tau))}\right) \cdot \text{poly}(r, p), \tag{3.5}$$

*where $\widetilde{\text{pred}}_{\mathcal{T}} \triangleq \mathbb{E}[\text{pred}_{\mathcal{T}}]$ is the expectation of the prediction matrix when achieving the minimum of the loss in Equation 3.3 with $\tau \in [0, 1]$, $\tilde{\lambda}_1(\tau), \tilde{\lambda}_d(\tau)$ and $\tilde{\lambda}_{d+1}(\tau)$ are the 1-st, d-th and $d + 1$-th eigenvalues of $\tilde{A} \triangleq \mathbb{E}\left[A - \frac{\tau|\tilde{E}|}{N^2} 11^\top\right]$, respectively.*

*Furthermore, the target error can be bounded by*

$$\mathbb{P}_{x \sim \mathcal{T}}(\text{pred}(x) \neq y_x) \leq O\left(\frac{(\tilde{\lambda}_1(\tau))^2}{\eta^4(\tilde{\lambda}_d(\tau) - \tilde{\lambda}_{d+1}(\tau))^2 \cdot n}\right) \cdot \text{poly}(r, p), \tag{3.6}$$

*where $\text{poly}(r, p)$ denotes a polynomial function of $r$ and $p$.*

In the following, we show that the introduction of $\tau$ through zero-mean regularization can tighten the error bound:

**Proposition 3.2.** *In the setting of Theorem 3.1, if $\tau \leq \frac{\tilde{\lambda}_1(0) - \tilde{\lambda}_2(0)}{\tilde{\lambda}_1(0)} = \frac{n \cdot \min\{p\alpha + r(p-1)\gamma, r\beta + p(r-1)\gamma\}}{\tilde{\lambda}_1(0)}$, then, with probability at least $1 - n^{-\zeta}$, we have*

$$\mathbb{P}_{x \sim \mathcal{T}}(\text{pred}(x) \neq y_x) \leq O\left(\frac{(1 - \tau)^2(\tilde{\lambda}_1(0))^2}{\eta^4(\tilde{\lambda}_d(0) - \tilde{\lambda}_{d+1}(0))^2 \cdot n}\right) \cdot \text{poly}(r, p), \tag{3.7}$$

*where $\tilde{\lambda}_1(0) = n\rho + n(r - 1)\beta + n(p - 1)\alpha + n(p - 1)(r - 1)\gamma$ and $\tilde{\lambda}_d(0) - \tilde{\lambda}_{d+1}(0) = n \cdot \min\{r(\beta - \gamma), p(\alpha - \gamma)\}$.*

As can be seen, the error decreases when the number of samples $n$ increases, and it also is controlled by the difference between the $d$-th and $d + 1$-th eigenvalues. This difference, in turn, depends on the gap between the across-class/across-domain connectivities $\beta$, $\alpha$ and the across-both connectivities $\gamma$. Furthermore, Proposition 3.2 provides additional insight that the incorporation of $\tau$ can tighten the error bound in the form of multiplying a factor of $(1 - \tau)^2$. This theoretical foundation underpins the proposed zero-mean regularization in the UDA task.

## 3.3 SUPERVISED CLASSIFICATION WITH NOISY LABELS

The primary objective of this subsection is to establish the supervised version of SpeCL with zero-mean regularization. We subsequently derive the closed-form optimal representations by minimizing the supervised SpeCL, which leads to a result similar to the neural collapse solution. Furthermore, we demonstrate that zero-mean regularization can mitigate label noise by implicitly reducing mislabeled weights in the noise transition matrix.

### 3.3.1 SUPERVISED SPECTRAL CONTRASTIVE LOSS

Similar to the supervised contrastive loss [26], we can define the supervised version of SpeCL with the zero-mean regularization for the labeled dataset $(X, Y)$ as

$$\mathcal{L}^{sup}(f; \tau) = \mathcal{L}^{sup}_{pos}(f) + \mathcal{L}_{neg}(f; \tau),$$

$$\mathcal{L}^{sup}_{pos}(f) = -\frac{2}{rn^2} \sum_{i=1}^n \sum_{j=1}^n \sum_{c=1}^r \mathbb{E}_{\substack{x \sim \mathcal{A}(x_{i,c}) \\ x^+ \sim \mathcal{A}(x_{j,c})}} \left[f(x)^\top f(x^+)\right], \text{ and}$$

$$\mathcal{L}_{neg}(f; \tau) = \frac{1}{r^2 n^2} \sum_{i=1}^n \sum_{j=1}^n \sum_{c=1}^r \sum_{k=1}^r \mathbb{E}_{\substack{x \sim \mathcal{A}(x_{i,c}) \\ x^- \sim \mathcal{A}(x_{j,k})}} \left[\left(f(x)^\top f(x^-) + \tau\right)^2\right],$$

$$\tag{3.8}$$

where $x_{i,c}$ denotes the $i$-th example in the class $c$, $n$ is the number of examples in each class, and $\mathcal{A}(x)$ denotes the augmentations of example $x$.

**Remark.** Note that we retain the condition $c = k$ within the negative part $\mathcal{L}_{neg}$, which is raised for two main reasons. Firstly, this condition aligns with the self-supervised form of spectral contrastive loss in Eq. 7, which facilitates the derivation of the positive and negative parts into a spectral decomposition as evidenced in the proof of Theorem 3.3; Secondly, the inclusion of the constraint $c \neq k$ in $\mathcal{L}_{neg}$ would result in that the supervised loss goes to infinity as the similarity $f(x)^\top f(x^+)$ in $\mathcal{L}_{pos}$ has the potential to become arbitrarily large. By incorporating positive pairs in $\mathcal{L}_{neg}$[3], we can effectively address this issue. This mechanism is analogous to the presence of numerator in the denominator in CE and InfoNCE, serving to ensure training stability.

The supervised version $\mathcal{L}^{sup}(f;\tau)$ differs from the self-supervised version $\mathcal{L}(f;\tau)$ in that $\mathcal{L}^{sup}_{pos}(f)$ uses samples in the same class as positive pairs, resulting in class-dependent connections in the positive pair graph for supervised spectral contrastive learning.

**Theorem 3.3.** *The global minimum of the supervised SpeCL $\mathcal{L}^{sup}(f;\tau)$ in Equation 3.8 is uniquely obtained at $\forall i \in [n], \forall c \in [r], \forall x \sim \mathcal{A}(x_{i,c}), f(x) = \hat{h}_c$, where $\hat{H} = [\hat{h}_1, ..., \hat{h}_r]^\top$ is the minimizer of $\left\| (r\mathrm{I} - \tau 11^\top) - H^\top H \right\|_F^2$.*

A more specific conclusion of Theorem 3.3 is that we have $\hat{H}^\top \hat{H} = r\mathrm{I} - \tau 11^\top$ when $r \leq d + 1$, and when $r > d + 1$, $\hat{H}^\top \hat{H} = \mathcal{P}_d(r\mathrm{I} - \tau 11^\top)$ where $\mathcal{P}_d(X)$ denotes the best $d$-rank approximation of $X$. These results provide insights into the structure of the learned representations and highlight the impact of zero-mean regularization on the geometric characteristics of the learned representations. One straightforward observation is that, as $\tau$ increases, the angle between representations of different classes (*i.e*, $\angle(\hat{h}_i, \hat{h}_j) = \arccos \frac{-\tau}{r-\tau}$ for $i \neq j$) increases, thereby leading to better discriminativeness. When $\tau = 1$ and $r \leq d + 1$, we have $\hat{H}^\top \hat{H} = (r-1)(\mathrm{I} - \frac{1}{r}11^\top)$ that forms the neural collapse solution[4] proposed in [39].

**Remark.** Resembling the neural collapse phenomenon [39], the optimal minimizer of supervised SpeCL is characterized by two manifestation for last-layer features when $r \leq d + 1$ and $\tau \in [0, 1]$: (i) *Within-class variability collapse*. The within-class variation of features becomes negligible as they collapse to their respective means. Specifically, for all classes $c$ and samples $x \sim \mathcal{A}(x_{i,c})$, the feature becomes a constant value $f(x) = \hat{h}_c$, leading to a reduction in with-class variability; (ii) *Convergence to an equiangular frame*. The vectors of the class means converge to having equal length, forming equal-sized angles between any pair. This is represented by the equation $H^\top H = r\mathrm{I} - \tau 11^\top$. These two manifestation indicates that the zero-regularized SpeCL exhibits a phenomenon similar to neural collapse even though it exclusively employs features without the linear classifier to calculate loss. Regarding the downstream linear classification, this suggests that *we may easily utilize class-mean features as the classifier*. Compared Table 1 to Table 12, class-mean features as classifier achieve comparable results with the trained classifier on CIFAR-10 and SVHN.

### 3.3.2 THE IMPLICATION OF MITIGATING LABEL NOISE

As mentioned in Section 3.3, the introduction of $\tau$ can encourage the discriminativeness of different classes. In the following, we will prove that zero-mean regularization can mitigate label noise. Consider the label noise transition matrix $W = (w_{ij})_{r \times r}$, where $w_{ij}$ denotes the probability of flipping the true class $j$ into class $i$. Then, for each sample $x_{i,y_x}$ in the true class $y_x \in [r]$, the probability of its label being $\hat{y}_x \in [r]$ is $w_{y_x \hat{y}_x}$. To analyze the supervised SpeCL in the presence of label noise, we let $\mathcal{L}^{sup}_{noise}(f;\tau)$ denote the expected loss on the label noise model, *i.e.*, $\mathcal{L}^{sup}_{noise}(f;\tau) = \mathbb{E}_{\hat{Y}|Y} \mathcal{L}^{sup}(f;\tau,(X,\hat{Y}))$, where $\hat{Y}$ is the flipped labels of $Y$. We can derive the results similar to Theorem 3.3 as follows:

**Theorem 3.4.** *For the noise transition matrix $W$ defined above, the global minimum of $\mathcal{L}^{sup}_{noise}(f;\tau)$ is uniquely obtained at $\forall i \in [n], \forall c \in [r], \forall x \sim \mathcal{A}(x_{i,c}), f(x) = \hat{h}_c$, where $\hat{H} = [\hat{h}_1, ..., \hat{h}_r]^\top$ is the minimizer of $\left\| (rW - \tau 11^\top) - H^\top H \right\|_F^2$.*

---

[3] We have $-2s \cdot f(x)^\top f(x^+) + \left[ f(x)^\top f(x^+) + \tau \right]^2 = \left[ f(x)^\top f(x^+) - (s - \tau)^2 \right] + const.$

[4] Our analysis covers all choices of representation dimension and the number of classes, unlike previous work [17, 36, 50] that only considers cases where the feature dimension is larger than the number of classes.

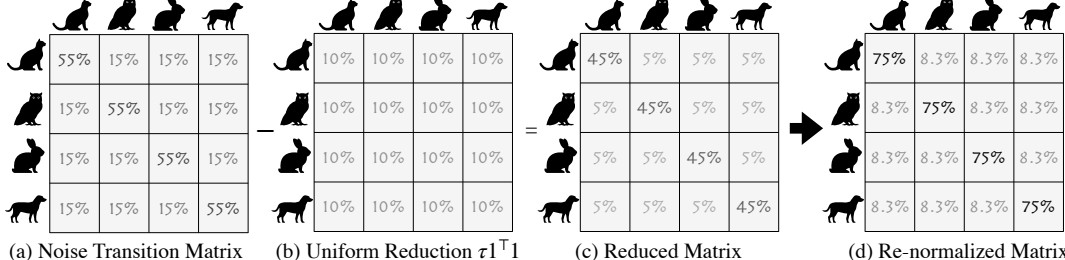

(a) Noise Transition Matrix     (b) Uniform Reduction $\tau 1^\top 1$     (c) Reduced Matrix     (d) Re-normalized Matrix

Figure 3: Illustration of zero-mean regularization for mitigating label noise. (a) The noise transition matrix that is clean-label dominant [63] with the noise rate 45%. (b) Zero-mean regularization causes all elements of the transition matrix to be subtracted by a constant $\tau = 10\%$. (c) The mislabeled (off-diagonal) weights are reduced to be 5% and the clean (on-diagonal) weights are reduced to 45%. (d) The re-normalized matrix shows that the overall noise rate is reduced to 25%, suggesting that zero-mean regularization is a simple yet effective way to mitigate label noise.

Table 1: Top-1 liner probing (%) and Top-1 validation accuracy (%) of self-supervised learning and supervised learning with the spectral contrastive loss in Equation B.1, respectively. All results reported by "mean $\pm$ std" are ran 3 trials. The results of the spectral contrastive loss with zero mean regularization $\tau > 0$ with better performance than the original spectral contrastive loss $\tau = 0$ are **highlighted**. The best results are underlined.

| Method | Self-Supervised Learning | | | Supervised Learning | | |
|---|---|---|---|---|---|---|
| | **CIFAR-10** | **CIFAR-100** | **SVHN** | **CIFAR-10** | **CIFAR-100** | **SVHN** |
| SpeCL | $86.22 \pm 0.12$ | $52.66 \pm 0.16$ | $89.77 \pm 0.19$ | $93.47 \pm 0.46$ | $65.59 \pm 0.75$ | $96.05 \pm 0.16$ |
| $\tau = 0.1$ | $\mathbf{86.83 \pm 0.12}$ | $\mathbf{53.97 \pm 0.19}$ | $\mathbf{90.26 \pm 0.07}$ | $\mathbf{94.21 \pm 0.13}$ | $\mathbf{68.15 \pm 0.19}$ | $\mathbf{96.22 \pm 0.05}$ |
| $\tau = 0.2$ | $\mathbf{86.84 \pm 0.10}$ | $\mathbf{55.27 \pm 0.02}$ | $\mathbf{90.78 \pm 0.07}$ | $\underline{\mathbf{94.52 \pm 0.05}}$ | $\mathbf{68.04 \pm 0.18}$ | $\underline{\mathbf{96.28 \pm 0.05}}$ |
| $\tau = 0.5$ | $\mathbf{87.15 \pm 0.12}$ | $\mathbf{56.37 \pm 0.27}$ | $\mathbf{91.10 \pm 0.13}$ | $\mathbf{94.36 \pm 0.12}$ | $\mathbf{69.06 \pm 0.34}$ | $\mathbf{96.22 \pm 0.09}$ |
| $\tau = 1.0$ | $\mathbf{87.09 \pm 0.16}$ | $\underline{\mathbf{56.63 \pm 0.10}}$ | $\underline{\mathbf{91.24 \pm 0.09}}$ | $\mathbf{94.38 \pm 0.06}$ | $\underline{\mathbf{69.59 \pm 0.11}}$ | $\mathbf{96.25 \pm 0.10}$ |

This theorem reveals that the immediate effect of $\tau$ is to implicitly reduce the mislabeled weights ($w_{ij}$, $i \neq j$) in the noise transition matrix, which in turn mitigates label noise. For a better understanding, we provide a more direct view for symmetric label noise.

**Proposition 3.5.** *Considering the symmetric label noise [15] in which $w_{cc'} = 1 - (r-1)\omega$ for $c = c'$, $w_{cc'} = \omega$ for $c \neq c'$, and $\omega < \frac{1}{r}$. If $\tau \geq r\omega$, let $\hat{f} = \arg\min_f \mathcal{L}_{noise}^{sup}(f; \tau)$, then $\frac{1}{\sqrt{1-r\omega}} \cdot \hat{f}$ is also the global minimizer of $\mathcal{L}^{sup}\left(f; \frac{\tau - r\omega}{1-r\omega}\right)$.*

Proposition 3.5 indicates that, when $\tau \geq r\omega$, using $\mathcal{L}_{noise}^{sup}(f; \tau)$ training on symmetric noisy labels is equivalent to using $\mathcal{L}^{sup}\left(f; \frac{\tau - r\omega}{1-r\omega}\right)$ training on clean labels. As illustrated in Figure 3, if the uniform reduction is $\tau = 15\%$, the re-normalized matrix will be an identify matrix.

## 4 EXPERIMENTS

In this section, we provide extensive experiments on the tasks of contrastive learning, supervised classification, unsupervised domain adaptation, and learning with noisy labels to verify the effectiveness of zero-mean regularization on several benchmark datasets with ResNets [21, 22]. **More experimental results and details can be found in the Appendix B.**

**Results of Self-Supervised Learning and Supervised Learning.** Table 1 reported the linear probing accuracy of self-supervised learned models and the Top-1 accuracy of supervised learned models with spectral contrastive loss in Equation 3.1 and supervised spectral contrastive loss in Equation 3.8. The results show that zero-mean regularization can learn more discriminative representations.

**Results of Unsupervised Domain Adaptation.** We evaluate zero-mean regularization across four digits datasets with different domains: SVHN (**S**) [37], MNIST (**M**) [28], USPS (**U**) [25], and MNIST-M (**M-M**) [14]. We first pre-train models with the modified spectral contrastive loss in Equation 3.1, and then fine-tune a linear classifier composited the models on the source domain.

Table 2: Top-1 accuracy (%) of unsupervised domain adaptation on four digit datasets. The results of the spectral contrastive loss with zero mean regularization $\tau > 0$ with better performance than the original spectral contrastive loss $\tau = 0$ are **highlighted**. The best results are underlined.

|  | S→M | S→U | S→M-M | M→ U | M→ M-M | M-M→ S | Average |
|---|---|---|---|---|---|---|---|
| Vanilla SpeCL | 88.10 | 80.37 | 75.24 | 94.47 | 51.19 | 37.32 | 71.12 |
| $\tau = 0.1$ | **88.16** | **82.76** | **76.10** | **95.71** | **72.10** | **44.78** | **76.60** |
| $\tau = 0.2$ | 86.44 | **82.96** | **76.10** | **94.87** | **63.63** | **37.45** | **73.58** |
| $\tau = 0.5$ | **88.44** | **83.16** | **77.34** | 90.43 | **64.68** | **48.03** | **75.35** |
| $\tau = 1.0$ | **88.73** | **83.81** | **79.81** | **96.16** | **59.48** | **38.73** | **74.45** |

Table 3: Top-1 test accuracy (mean ±std, %) of supervised spectral contrastive loss with zero-mean regularization in Equation 3.8 on benchmark datasets with symmetric label noise. The results of the spectral contrastive loss with zero mean regularization $\tau > 0$ with better performance than the original spectral contrastive loss $\tau = 0$ are **highlighted**. The best results are underlined.

| Datasets | Methods | Symmetric Noise Rate | | | |
|---|---|---|---|---|---|
|  |  | 0.2 | 0.4 | 0.6 | 0.8 |
| CIFAR-10 | CE | $82.88 \pm 0.04$ | $68.99 \pm 0.15$ | $52.27 \pm 0.20$ | $51.62 \pm 0.08$ |
|  | Focal | $60.91 \pm 0.07$ | $46.73 \pm 0.13$ | $29.47 \pm 0.04$ | $13.54 \pm 0.03$ |
|  | GCE | $91.35 \pm 0.04$ | $89.14 \pm 0.01$ | $80.50 \pm 0.07$ | $52.25 \pm 0.03$ |
|  | Vanilla SpeCL | $90.80 \pm 0.01$ | $88.29 \pm 0.13$ | $85.87 \pm 0.07$ | $75.51 \pm 0.27$ |
|  | $\tau = 0.1$ | $\mathbf{91.18 \pm 0.07}$ | $\mathbf{88.99 \pm 0.07}$ | $\mathbf{86.66 \pm 0.04}$ | $\mathbf{80.29 \pm 0.39}$ |
|  | $\tau = 0.2$ | $\mathbf{91.57 \pm 0.03}$ | $\mathbf{89.13 \pm 0.12}$ | $\mathbf{86.81 \pm 0.03}$ | $\mathbf{80.06 \pm 0.39}$ |
|  | $\tau = 0.5$ | $\mathbf{91.31 \pm 0.06}$ | $\mathbf{89.49 \pm 0.02}$ | $\mathbf{86.59 \pm 0.09}$ | $\mathbf{78.51 \pm 0.25}$ |
|  | $\tau = 1.0$ | $\mathbf{91.42 \pm 0.07}$ | $\mathbf{89.26 \pm 0.05}$ | $\mathbf{86.98 \pm 0.07}$ | $\mathbf{76.63 \pm 0.30}$ |
| CIFAR-100 | CE | $60.94 \pm 0.06$ | $46.07 \pm 0.01$ | $30.33 \pm 0.02$ | $14.41 \pm 0.06$ |
|  | Focal | $60.91 \pm 0.07$ | $46.73 \pm 0.13$ | $29.47 \pm 0.04$ | $13.54 \pm 0.03$ |
|  | GCE | $68.91 \pm 0.03$ | $64.87 \pm 0.07$ | $56.04 \pm 0.04$ | $9.13 \pm 0.00$ |
|  | Vanilla SpeCL | $67.87 \pm 0.05$ | $62.97 \pm 0.11$ | $57.51 \pm 0.02$ | $51.10 \pm 0.08$ |
|  | $\tau = 0.1$ | $\mathbf{68.63 \pm 0.09}$ | $\mathbf{64.46 \pm 0.04}$ | $\mathbf{59.41 \pm 0.06}$ | $\mathbf{52.32 \pm 0.09}$ |
|  | $\tau = 0.2$ | $\mathbf{69.60 \pm 0.07}$ | $\mathbf{65.04 \pm 0.07}$ | $\mathbf{59.46 \pm 0.04}$ | $\mathbf{53.90 \pm 0.06}$ |
|  | $\tau = 0.5$ | $\mathbf{69.57 \pm 0.06}$ | $\mathbf{65.05 \pm 0.06}$ | $\mathbf{59.17 \pm 0.09}$ | $\mathbf{52.80 \pm 0.08}$ |
|  | $\tau = 1.0$ | $\mathbf{69.50 \pm 0.15}$ | $\mathbf{65.37 \pm 0.07}$ | $\mathbf{59.41 \pm 0.01}$ | $\mathbf{52.60 \pm 0.07}$ |

Table 2 demonstrates that zero-mean regularization ($\tau > 0$) can help decoupling class information of representations across domains. Experiments on DomainNet [41] are provided in Appendix B.4.

**Results of Learning with Noisy Labels.** To further verify that zero-mean regularization can mitigate label noise. We conduct experiments in the scenario of symmetric label noise, considering baselines CE, Focal loss [32], and GCE [59]. As shown in Table 3, $\tau > 0$ performs better than SpeCL in all cases, especially for high noise. This also show that contrastive learning can mitigate label noise.

## 5 CONCLUSION

This paper introduces zero-mean regularization to enhance spectral contrastive loss, which equivalently incorporates an additive factor into the loss component involving negative pairs and thus makes representations of negative pairs opposite. Provable accuracy guarantees are achieved under linear probe evaluation for contrastive learning with restricted model classes. Additionally, zero-mean regularization reduces positive-pair weights uniformly during contrastive pretraining for unsupervised domain adaptation, tightening downstream error on target domains. The supervised version of spectral contrastive loss reveals a structure resembling the neural collapse phenomenon, with larger regularization strength indicating improved discriminativeness. Moreover, zero-mean regularization mitigates label noise by implicitly reducing mislabeled weights in the noise transition matrix. These findings highlight the superiority of zero-mean regularization in enhancing contrastive learning and its potential applications in various domains.

## ACKNOWLEDGE

This work was supported in part by National Natural Science Foundation of China under Grants 92270116, 62071155, and 623B2031, and in part by the Fundamental Research Funds for the Central Universities (Grant No.HIT.OCEF.2023043 and HIT.DZJJ.2023075).

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

# Appendix for "Zero-Mean Regularized Sepctral Contrastive Learning"

## A    MORE CLARIFICATION

### A.1    "WEIGHT REDUCTION IN POSITIVE PAIRS" BEYOND ZERO-MEAN REGULARIZATION

The main idea of zero-mean regularization is to implicitly reduce the weights of wrong connections in the positive-pair graph, as stated throughout the paper. This idea is fully depicted in the unsupervised domain adaptation (see Eq. 3.3) and learning with noisy labels (see Theorem 3.4) in Section 3.3.2. In this paper, we focus on the spectral contrastive loss, since it is theoretically sound and easier to analyze. The final manifestation is accordingly expressed in the concise term of the loss-specific *zero-mean regularization*.

We further attempt to apply the core idea of "weight reduction in positive pairs" to the InfoNCE loss that is widely used in many popular contrastive learning schemes, such as SimCLR [6], MoCo [23], and CPC [38]. Specifically, we modify the InfoNCE loss by uniformly reducing the importance of positive pair as follows:

$$
\begin{aligned}
&- \log \frac{(1 - \tau) e^{sim(x, x^+)}}{(1 - \tau) e^{sim(x, x^+)} + \sum_{i=1}^{k} e^{sim(x, x_i^-)}} \\
&= - \log \frac{e^{sim(x, x^+) - \tau'}}{e^{sim(x, x^+) - \tau'} + \sum_{i=1}^{k} e^{sim(x, x_i^-)}} \\
&= - \log \frac{e^{sim(x, x^+)}}{e^{sim(x, x^+)} + \sum_{i=1}^{k} e^{sim(x, x_i^-) + \tau'}},
\end{aligned}
\tag{A.1}
$$

where $sim(x_1, x_2)$ denotes the similarity between $x_1$ and $x_2$, $\tau' = \log \frac{1}{1 - \tau} > 0$ and $\tau \in [0, 1)$. As can be seen, the derived form that adds a margin term is similar to margin-based losses, particularly the negative-margin softmax loss [33]. From this point of view, weight reduction in positive pairs coincides with the motivation of margin-based losses [12, 33, 34, 54, 61] that enlarges the discriminativeness with intuitive decision boundaries.

To empirically validate the efficacy of the modified term InfoNCE loss in Equation A.1, we conduct experiments on self-supervised learning and supervised learning in Table 4. In these experiments, we utilize CIFAR-10/-100 and SVHN datasets as benchmarks. We believe that these additional experiments not only provide further evidence of the versatility embedded within the concept of "weight reduction in wrong positive pairs" but also highlight its potential to enhance the performance of contrastive learning algorithms on real-world datasets.

Table 4: Top-1 liner probing (%) and Top-1 validation accuracy (%) of self-supervised learning and supervised learning with the modified InfoNCE in Equation A.1, respectively. The results of the InfoNCE loss with the margin $\tau > 0$ obtaining better performance than the original spectral contrastive loss $\tau = 0$ are **highlighted**. The best results are underlined.

| Method | Self-Supervised Learning | | | Supervised Learning | | |
|---|---|---|---|---|---|---|
| | **CIFAR-10** | **CIFAR-100** | **SVHN** | **CIFAR-10** | **CIFAR-100** | **SVHN** |
| InfoNCE | 86.56 | 55.51 | 90.77 | 94.18 | 71.43 | 96.11 |
| $\tau = 0.05$ | **86.64** | **55.68** | **91.05** | 94.04 | **71.80** | **96.24** |
| $\tau = 0.1$ | **86.97** | **56.07** | **90.93** | **94.27** | 71.56 | **96.17** |
| $\tau = 0.2$ | **86.65** | **55.55** | 90.77 | **94.19** | 72.07 | **96.20** |
| $\tau = 0.5$ | **87.32** | **56.01** | **90.95** | 94.17 | **72.44** | **96.18** |
| $\tau = 0.9$ | 86.48 | **55.88** | **90.95** | **94.43** | 72.29 | **96.16** |

### A.2    DOWNSTREAM ERROR BOUND FOR CONTRASTIVE LEARNING

There are several works focusing on the SpeCL [18, 19, 20], which provide provable accuracy guarantees under linear probe evaluation in context of an implicit $m$-way partition of $\mathcal{X}$, *i.e.*, they are

disjoint non-empty sub-graphs of $\mathcal{X}$ such that $\mathcal{X} = \cup_{i \in [m]} \mathcal{S}_i$ ($\forall x \in \mathcal{X}$, let $\mathrm{id}_x$ be the index such that $x \in \mathcal{S}_{\mathrm{id}_x}$). Specifically, HaoChen et al. [19] proved the efficacy of SpeCL under the assumption that the representation dimension $d$ exceeds the number of sub-graphs $m$. Subsequently, HaoChen and Ma [18] extended these results by studying representations learned within a constrained model class $\mathcal{F}$ while imposing the condition $d = m$. In this work, we prove a downstream error bound when $m > d$ as follows:

**Theorem A.1.** *Let $\mathcal{F}$ be a class of functions of $\mathcal{X} \to \mathbb{R}^d$ and let $\hat{f} = \arg\min_{f \in \mathcal{F}} \mathcal{L}(f; \tau)$ be the minimizer of the SpeCL. Assume that:*

1. *($\epsilon$-separability). The probability of a positive pair belonging to two different sets is less than $\epsilon$, that is, $\mathrm{Pr}_{(x,x^+) \sim p_{\mathrm{pos}}}(\mathrm{id}_x \neq \mathrm{id}_{x^+}) \leq \epsilon$;*

2. *(Alignment) The downstream label $y(x)$ is a constant on each $\mathcal{S}_i$ for $i \in [m]$;*

3. *($\mathcal{F}$-implementable inner-cluster connection larger than $\delta$). For any $f \in \mathcal{F}$ and any linear head $w \in \mathbb{R}^d$, let function $g(x) = w^\top f(x)$. For any $i \in [m]$, we have $Q_{\mathcal{S}_i}(g) :=$ $\dfrac{\mathbb{E}_{(x,x^+) \sim p_{\mathrm{pos}}^{\mathcal{S}_i}}\left[(g(x)-g(x^+))^2\right]}{\mathbb{E}_{x \sim p_{\mathrm{data}}^{\mathcal{S}_i}, x' \sim p_{\mathrm{data}}^{\mathcal{S}_i}}\left[(g(x)-g(x'))^2\right]} \geq \delta$;*

4. *(Implementability). There exists a function $f \in \mathcal{F}$ such that $f(x) = v_{id_x}$ for all $x \sim p_{\mathrm{data}}$, where $\{v_1, v_2, ..., v_m\}$ is a set of different vectors that achieve the global minimum of $\|\sum_{i=1}^m p_i v_i v_i^\top - \mathrm{I}\|_F^2 + 2\tau \cdot \|\sum_{i=1}^m p_i v_i\|_2^2$, and $p_i = \mathrm{Pr}_{x \sim p_{\mathrm{data}}}(x \in \mathcal{S}_i)$.*

*Let $p_{\min} = \min_{i \in [m]} \mathrm{Pr}_{x \sim p_{\mathrm{data}}}(x \in \mathcal{S}_i)$ and $\Delta = \max_{i,j} \|v_i - v_j\|_2^2$. For $m > d$, if $\epsilon\Delta < 1$, then there exists a linear head $W \in \mathbb{R}^{d \times d}$ which achieves the following downstream error*

$$\mathbb{E}_{x \sim p_{\mathrm{data}}}\left[\left\|W\hat{f}(x) - v_{y(x)}\right\|_2^2\right] \leq \frac{\epsilon\Delta(1 + \sqrt{\epsilon\Delta})p_{\min}}{2\delta(p_{\min} - \epsilon)}. \tag{A.2}$$

**Remark.** In Theorem A.1, we assume that the model class $\mathcal{F}$ is implementable and capable of expressing some vectors $V = [v_1, ..., v_m]$ that achieve the global minimum of $\|\sum_{i=1}^m p_i v_i v_i^\top - \mathrm{I}\|_F^2 + 2\tau \cdot \|\sum_{i=1}^m p_i v_i\|_2^2$ (see Lemma A.3 in the appendix for the specific form of $V$). In [18], these vectors are concretely specified as one-hot vectors, which actually corresponds to the case of $\tau = 0$. Otherwise, our error bound under linear probing is characterized as $\mathbb{E}_{x \sim p_{\mathrm{data}}} \mathbb{I}(y(x) \neq \arg\min_c \|W\hat{f}(x) - v_c\|_2^2)$ and can be further bounded by the error $\mathbb{E}_{x \sim p_{\mathrm{data}}}[\|W\hat{f}(x) - v_{y(x)}\|_2^2]$. Moreover, the right-hand side in Equation A.2 can be further tightened once we choose vectors that minimize $\Delta = \max_{i \neq j} \|v_i - v_j\|_2^2$. Therefore, our main result in Theorem A.1 provides a general bound for spectral contrastive learning when $m > d$, which extends the prior works and completes the picture of spectral contrastive learning.

Importantly, Theorem A.1 also reveals a trade-off between inter-partition separability and intra-partition compactness. Intuitively, good inter-partition separability involves maximizing the distance between any two distinct partitions, *i.e.*, $\max_V \min_{i \neq j} \|v_i - v_j\|_2^2$, while the notion of intra-partition compactness anticipates the reduction in $\Delta = \max_{i,j} \|v_i - v_j\|_2^2$, contributing to a more stringent bound on the right-hand side of Eq. 3.3. The intricate interdependence between $\max_V \min_{i \neq j} \|v_i - v_j\|_2^2$ and $\min_V \max_{i \neq j} \|v_i - v_j\|_2^2$ elucidates the nuanced nature of the trade-off in the understanding of self-supervised learning.

### A.3 COMPARISON TO CONTRASTIVE LAPLACIAN EIGENMAPS

While there are similarities between Zero-mean regularized spectral loss (Zero-SpeCL) in Eq. A.2 and contrastive Laplacian eigenmaps (COLES) [64], there are distinct differences that are crucial to highlight:

- **Different Negative Components**. The contrastive objective (to be maximized) can generally formulated as the objective $J(f) = \mathbb{E}_{(x,x^+) \sim p_{\mathrm{pos}}} s(f(x), f(x^+)) + \eta \mathbb{E}_{(x,x^-) \sim p_{\mathrm{data}}} \tilde{s}(f(x), f(x^-))$. While the positive components of Zero-SpeCL and COLES are the same, their negative components differ. The negative component of COLES is $\tilde{s}(f(x), f(x^-)) = -f(x)^\top f(x^-)$, while Zero-SpeCL is $\tilde{s}(f(x), f(x^-)) = -(f(x)^\top f(x^-) + \tau)^2$.

- **Different Reasons for Covariance Term**. Both Zero-SpeCL and COLES introduce the term $\|\mathbb{E}_x f(x)f(x)^\top - I\|_F^2$. The motivation of COLES is to softly satisfy the constraint $F^\top F = I$ that removes an arbitrary scaling factor in the embeddings [3]. In addition, it also helps avoid collapsed solutions, since without the constraint $F^\top F = I$, the derived graph Dirichlet energy $-Tr(F^\top \Delta W F)$ will be minimized when all representations collapse to a constant vector. In contrast, the covariance term $\mathcal{R}_1(f) = \|\mathbb{E}_x f(x)f(x)^\top - I\|_F^2$ in Zero-SpeCL appears as a part of an equivalent form in Eq. 3.2.

- **Different Overall Objective.** COLES represents a constrained graph Dirichlet energy minimization problem (contrastive Laplacian eigenmaps), i.e., $\min_{F^\top F = I} -Tr(F^\top \Delta W F)$. On the other hand, Zero-SpeCL depicts the low-rank matrix approximation problem $\min_F |A - F^\top F|_F^2$ (as shown in Eq. 3.3).

## A.4 THE ROLE OF $\tau$

In this paper, we propose setting $\tau \in [0, 1]$ to balance the regularization strength of zero-mean regularization. Although in Section 3, we do not explicitly constrain $\tau \leq 1$, theoretical and empirical evidence supports the necessity of such a constraint:

- In Theorem 3.1, we assume that $\tau < \frac{\tilde{\lambda}_1(0) - \tilde{\lambda}_d(0)}{\tilde{\lambda}_1(0)} < 1$ to ensure that the introduction of $\tau$ will not alter the value of the $d$-th eigenvalue in Eq. (C.23). In Proposition 3.2, we assume that $\tau \leq \frac{\tilde{\lambda}_1(0) - \tilde{\lambda}_2(0)}{\tilde{\lambda}_1(0)} < \frac{\tilde{\lambda}_1(0) - \tilde{\lambda}_d(0)}{\tilde{\lambda}_1(0)} < 1$ to guarantee that the first eigenvalue of $\tilde{A}'$ is $\tilde{\lambda}_1(\tau) = (1 - \tau)\tilde{\lambda}_1(0)$ (as can be seen, $\tau \leq 1$ is essential to avoid the presence of negative eigenvalues of $\tilde{A}'$) in Eq. (C.32), thus facilitating a more intuitive comparison with the bound at $\tau = 0$. While it is possible to draw conclusions within the range of $[\frac{\tilde{\lambda}_1(0) - \tilde{\lambda}_d(0)}{\tilde{\lambda}_1(0)}, 1]$, it requires a more nuanced comparison of the order of these eigenvalues, predictably resulting in an obviously different bound than at $\tau = 0$.

- In Theorem 3.3, we prove that the global minimum of the supervised spectral contrastive loss is achieved at $\hat{H} = \arg\min_H \|(rI - \tau 11^\top) - H^\top H\|_F^2$. When $r \leq d + 1$, $\|(rI - \tau 11^\top) - H^\top H\|_F^2$ has a minimum of zero only for $\tau \in [0, 1]$, corresponding to $\hat{H}^\top \hat{H} = rI - \tau 11^\top$, where the angle between representations of different classes is $\arccos \frac{-\tau}{r - \tau} > 90°$. However, for $\tau > 1$, there exists no $\hat{H}$ that satisfies $\hat{H}^\top \hat{H} = rI - \tau 11^\top$, as it would lead to a paradoxical situation where $0 \leq 1^\top \hat{H}^\top \hat{H} 1 = 1^\top (rI - \tau 11^\top)1 = r^2(1 - \tau) < 0$.

- We conducted experiments for $\tau > 1$ as presented in Tables 9 and 11 of the Appendix. As observed, consistently better results are obtained with $\tau \in [0, 1]$, while excessively large $\tau$ values hinder overall performance. Thus, we suggest limiting $\tau$ to the range of $[0, 1]$ to balance the regularization and discriminative capabilities.

# B MORE EXPERIMENTAL DETAILS AND RESULTS

In this section, we present the training details and experimental results on several tasks, including self-supervised contrastive learning, supervised contrastive learning, unsupervised domain adaptation, and learning with noisy labels.

## B.1 DATASETS

**Small Datasets.** In our experiments, we consider several commonly-used datasets, namely CIFAR-10/-100 [27], SVHN (**S**) [37], MNIST (**M**) [28], USPS (**U**) [25], and MNIST-M (**M-M**) [14]. For the tasks of self-supervised contrastive learning and supervised classification, we utilize CIFAR-10/-100 and SVHN datasets. These datasets are also employed for the evaluation of learning with noisy labels. To investigate unsupervised domain adaptation, we consider four domains of digit datasets: SVHN, MNIST, USPS, and MNIST-M. For small datasets, each image is resized to a standardized size of $32 \times 32$ with 3 color channels to ensure uniformity in input sizes.

Table 5: Top-1 linear probing of self-supervised learning with the spectral contrastive loss in Equation B.1 under various values of $\mu$ and $\tau$. Notably, the results obtained with the spectral contrastive loss incorporating zero mean regularization $\tau > 0$ and outperforming the baseline (the original spectral contrastive loss $\tau = 0$) are **highlighted**. The best results are underlined for clarity.

| Method | CIFAR-10 | CIFAR-100 | SVHN | CIFAR-10 | CIFAR-100 | SVHN |
| --- | --- | --- | --- | --- | --- | --- |
| | | $\mu = 1.0$ | | | $\mu = 5.0$ | |
| Vanilla SpeCL | 82.54 | 40.45 | 84.80 | 86.17 | 51.75 | 88.79 |
| $\tau = 0.1$ | **82.97** | **42.51** | **86.35** | **86.80** | **52.87** | **89.36** |
| $\tau = 0.2$ | **83.11** | **42.98** | **85.65** | **86.78** | **53.13** | **90.19** |
| $\tau = 0.5$ | **83.22** | **43.36** | **86.23** | **87.23** | **53.89** | **91.38** |
| $\tau = 1.0$ | **83.78** | **44.21** | **86.84** | **87.46** | **54.98** | **91.58** |

| Method | CIFAR-10 | CIFAR-100 | SVHN | CIFAR-10 | CIFAR-100 | SVHN |
| --- | --- | --- | --- | --- | --- | --- |
| | | $\mu = 10.0$ | | | $\mu = 20.0$ | |
| Vanilla SpeCL | 86.25 | 52.44 | 89.58 | 85.64 | 51.64 | 88.82 |
| $\tau = 0.1$ | **86.77** | **53.98** | **90.36** | 85.33 | **54.26** | **91.03** |
| $\tau = 0.2$ | **86.93** | **55.25** | **90.80** | 85.62 | **55.23** | **91.43** |
| $\tau = 0.5$ | **87.05** | **56.61** | **91.10** | **86.47** | **56.17** | **92.01** |
| $\tau = 1.0$ | **87.24** | **56.77** | **91.12** | 85.87 | **55.80** | **92.13** |

**ImageNet.** We also include a subset of the ImageNet dataset called ImageNet-100. ImageNet ILSVRC2012 [11] consists of approximately 1.2 million training images and 50,000 validation images, divided into 1,000 classes. However, due to the limited computing resources and time, we focus specifically on the first 100 classes of ImageNet, forming the ImageNet-100 dataset. Each image in this subset is resized to dimensions of $64 \times 64$ with 3 color channels.

**DomainNet.** For unsupervised domain adaptation, we employ the DomainNet dataset [41]. DomainNet is a large-scale dataset designed specifically for unsupervised domain adaptation tasks. It contains approximately 600,000 images distributed among 345 classes across six domains. These six domains included in DomainNet are clipart, painting, real, sketch, infograph, and quickdraw. In our experiments, we focus on four domains from the cleaned version of DomainNet, including clipart, painting, real, and sketch. Each image in these domains is resized to dimensions of $64 \times 64$ with 3 color channels to ensure consistency in input sizes.

## B.2 TRAINING DETAILS.

**Data Augmentation.** We adopt the same image augmentation settings as used in the SimSiam method [7]. These augmentation techniques include *RandomResizedCrop*, *ColorJitter*, *RandomGrayscale*, *GaussianBlur*, and *RandomHorizontalFlip*.

**Network Architectures.** For smaller datasets such as CIFAR-10/-100 and digit datasets (SVHN, MNIST, USPS, MNIST-M), we employ the PreAct-ResNet-18 architecture [22] as the backbone network. The projection layers consist of a 2-layer MLP with a hidden dimension of 2048 and an output dimension of 1024. For larger-scale datasets such as ImageNet and DomainNet, we utilize the ResNet-50 architecture [21] as the backbone network. The projection layers in this case also consist of a 2-layer MLP, with both the hidden dimension and the output dimension set to 4096.

**Optimization Details.** In this paper, we train these self-supervised (or supervised) models for a total of 200 epochs. The training process utilizes the SGD optimizer with a momentum of 0.9, a learning rate of 0.1, and a weight decay of 5e-4. To dynamically adjust the learning rate throughout the 200 epochs, we use the cosine decay learning rate schedule [35]. For datasets with smaller sizes, we set the batch size to 512 on 1 GPU, while for ImageNet-100 and DomainNet, the batch size was set to 1536 on 8 GPUs. All experiments were implemented using PyTorch and executed on NVIDIA GTX 2080Ti GPUs.

Table 6: Top-1 validation accuracy of supervised learning with the spectral contrastive loss in Equation 3.8 under various values of $\mu$ and $\tau$. Notably, the results obtained with the spectral contrastive loss incorporating zero mean regularization $\tau > 0$ and outperforming the baseline (the original spectral contrastive loss $\tau = 0$) are **highlighted**. The best results are underlined for clarity.

| Method | CIFAR-10 | CIFAR-100 | SVHN | CIFAR-10 | CIFAR-100 | SVHN |
| --- | --- | --- | --- | --- | --- | --- |
| | | $\mu = 1.0$ | | | $\mu = 5.0$ | |
| Vanilla SpeCL | 87.01 | 33.18 | 90.04 | 93.52 | 56.13 | 95.91 |
| $\tau = 0.1$ | **87.69** | **35.22** | **95.89** | **94.28** | **57.90** | **95.93** |
| $\tau = 0.2$ | **88.72** | **39.82** | **96.04** | **94.31** | **57.73** | **96.06** |
| $\tau = 0.5$ | **89.45** | **42.38** | **96.11** | **94.50** | **59.26** | **96.09** |
| $\tau = 1.0$ | **89.74** | **41.69** | **95.97** | **94.33** | **59.56** | **96.07** |
| Method | CIFAR-10 | CIFAR-100 | SVHN | CIFAR-10 | CIFAR-100 | SVHN |
| | | $\mu = 10.0$ | | | $\mu = 20.0$ | |
| Vanilla SpeCL | 94.05 | 65.68 | 95.89 | 94.14 | 70.70 | 96.12 |
| $\tau = 0.1$ | **94.39** | **68.11** | **96.21** | **94.28** | 70.50 | **96.16** |
| $\tau = 0.2$ | **94.59** | **68.30** | **96.23** | **94.30** | **71.03** | **96.25** |
| $\tau = 0.5$ | **94.47** | **68.74** | **96.14** | **94.64** | **71.04** | **96.22** |
| $\tau = 1.0$ | **94.43** | **69.74** | **96.31** | 94.06 | **71.94** | 95.92 |

**Implementation of Spectral Contrastive Loss.** The pseudocode for the spectral contrastive loss, as well as its supervised version in PyTorch-style, is provided in Table 7. It is worth noting that, similar to other methods such as SimSiam [7], BYOL [16], SimCLR [6], and others, we perform $\ell_2$-normalization on the final output to compute the loss. Additionally, followed HaoChen et al. [19], we introduce an extra hyper-parameter $\mu$ (the default value of $\mu$ is 10) to balance the positive and negative components of the loss.

## B.3 EXPERIMENTS ON CONTRASTIVE LEARNING AND SUPERVISED LEARNING

The empirical objective of spectral contrastive loss with zero-mean regularization used in this paper is formulated as

$$\mathcal{L}^{ssl}(f;\tau) = -\frac{2}{n}\sum_{i=1}^{n} \mathbb{E}_{\substack{x \sim \mathcal{A}(x_i) \\ x^+ \sim \mathcal{A}(x_i)}} [f(x)^\top f(x^+)] + \frac{1}{n^2}\sum_{i=1}^{n}\sum_{j=1}^{n} \mathbb{E}_{\substack{x \sim \mathcal{A}(x_i) \\ x^- \sim \mathcal{A}(x_j)}} [((f(x)^\top f(x^-)) + \tau)^2],$$

(B.1)

where $x_i$ denotes the $i$-th example in the input space, and $\mathcal{A}(x)$ denotes the augmentations of example $x$.

The supervised version of spectral contrastive loss with the zero-mean regularization for the labeled dataset $(\mathcal{X}, Y)$ as

$$\mathcal{L}^{sup}(f;\tau) = \mathcal{L}^{sup}_{pos}(f) + \mathcal{L}_{neg}(f;\tau),$$

$$\mathcal{L}^{sup}_{pos}(f) = -\frac{2}{rn^2}\sum_{i=1}^{n}\sum_{j=1}^{n}\sum_{c=1}^{r} \mathbb{E}_{\substack{x \sim \mathcal{A}(x_{i,c}) \\ x^+ \sim \mathcal{A}(x_{j,c})}} [f(x)^\top f(x^+)], \text{ and}$$

(B.2)

$$\mathcal{L}_{neg}(f;\tau) = \frac{1}{r^2 n^2}\sum_{i=1}^{n}\sum_{j=1}^{n}\sum_{c=1}^{r}\sum_{k=1}^{r} \mathbb{E}_{\substack{x \sim \mathcal{A}(x_{i,c}) \\ x^- \sim \mathcal{A}(x_{j,k})}} \left[ (f(x)^\top f(x^-) + \tau)^2 \right],$$

where $x_{i,c}$ denotes the $i$-th example in the class $c$, $n$ is the number of examples in each class, and $\mathcal{A}(\boldsymbol{x})$ denotes the augmentations of example $\boldsymbol{x}$.

**Experimental Details.** We conduct experiments on self-supervised and supervised spectral contrastive learning using PreAct-ResNet-18 models pre-trained on CIFAR-10, CIFAR-100, SVHN, and a subset of ImageNet comprising the first 100 classes. The training process involves training the networks for 200 epochs. To evaluate the performance of these models in linear classification, we employ an independent linear classifier on the fixed representations obtained during contrastive

Table 7: PyTorch-like pseudocode of spectral contrastive loss and its supervised version.

```python
# Spectral Contrastive Loss
class SpectralContrastiveLoss(nn.Module):
    def __init__(self, mu=1.0, tau=0.0):
        super().__init__()
        self.tau = tau
        self.mu = mu

    def forward(self, z1, z2, labels=None):
        # If labels is not None, the loss becomes supervised.
        assert z1.shape == z2.shape and len(z1.shape)==2
        device = z1.device
        N = z1.shape[0]
        z = torch.cat([z1, z2], dim=0)

        z = F.normalize(z, dim=1)
        logits = torch.matmul(z, z.T)

        # mask-out negative pairs
        if labels is None:
            mask = torch.eye(N, dtype=torch.float32, device=device).
                                                    repeat(2, 2)
        else:
            mask = torch.eq(labels, labels.T).float().repeat(2, 2)

        loss_part1 = -2 * (logits * mask).sum() / mask.sum()
        loss_part2 = self.mu * ((logits + self.tau) ** 2).mean()
        return loss_part1 + loss_part2
```

Table 8: Top-1 accuracy under linear probe evaluation (%) on ImageNet-100. The results of the spectral contrastive loss with zero mean regularization $\tau > 0$ with better performance than baseline (the original spectral contrastive loss $\tau = 0$) are **highlighted**. The best results are underlined.

| Method | Self-Supervised Learning | |
| --- | --- | --- |
| | ImageNet-100 | DomainNet |
| Vanilla SpeCL | 40.82 / 69.32 | 34.266 / 57.794 |
| $\tau = 0.1$ | 40.50 / **70.38** | **35.912 / 59.314** |
| $\tau = 0.2$ | **41.86 / 71.58** | **37.037 / 60.632** |
| $\tau = 0.5$ | **42.90 / 72.48** | **37.844 / 62.039** |
| $\tau = 1.0$ | 38.26 / 68.10 | **35.657 / 60.202** |

pre-training. This classifier is trained using labeled data. The reported results correspond to the top-1 accuracy achieved by these models.

**Experimental Results.** Table 5 and Table 6 present the Top-1 accuracy results for self-supervised learned models and supervised learned models, respectively. These results clearly demonstrates that zero-mean regularization consistently outperforms the Baseline (models without zero-mean regularization) across most scenarios. Notably, the best performance is consistently achieved when utilizing zero-mean regularization in all cases. Similar conclusions are also reflected in Table 8 which offers the results of self-supervised learning on ImageNet and DomainNet. Moreover, Table 9 reports the results obtained using a fixed value of $\mu = 10$ and different values of $\tau$, which shows that excessively large values of $\tau$ can hinder the improvement achieved by zero-mean regularization. Therefore, we suggest limiting the range of $\tau$ to $[0, 1]$.

Table 9: Top-1 test accuracy (mean $\pm$std, %) of spectral contrastive learning with a fixed value of $\mu = 10$ and various values of $\tau$ on benchmark datasets with symmetric label noise. Notably, the results obtained with the spectral contrastive loss incorporating zero mean regularization $\tau > 0$ and outperforming the baseline (the original spectral contrastive loss $\tau = 0$) are **highlighted**. The best results are underlined for clarity.

| Method | Self-Supervised Learning | | | Supervised Learning | | |
|---|---|---|---|---|---|---|
| | **CIFAR-10** | **CIFAR-100** | **SVHN** | **CIFAR-10** | **CIFAR-100** | **SVHN** |
| Vanilla SpeCL | 86.25 | 52.44 | 89.58 | 94.05 | 65.68 | 95.89 |
| $\tau = 0.1$ | **86.77** | **53.98** | **90.36** | **94.39** | **68.11** | **96.21** |
| $\tau = 0.2$ | **86.93** | **55.25** | **90.80** | **94.59** | **68.30** | **96.23** |
| $\tau = 0.5$ | **87.05** | **56.61** | **91.10** | **94.47** | **68.74** | **96.14** |
| $\tau = 1.0$ | **87.24** | **56.77** | **91.12** | **94.43** | **69.74** | **96.31** |
| $\tau = 5.0$ | 85.80 | **54.82** | **91.04** | 94.11 | **68.99** | **96.07** |
| $\tau = 10.0$ | 82.44 | 50.02 | 90.25 | 93.17 | **66.21** | 95.82 |
| $\tau = 20.0$ | 73.81 | 35.96 | 86.75 | 89.18 | 51.98 | 95.67 |

## B.4 EXPERIMENTS ON UNSUPERVISED DOMAIN ADAPTATION

We conduct empirical validation of unsupervised domain adaptation using spectral contrastive pretraining on four digit datasets: SVHN (**S**) [37], MNIST (**M**) [28], USPS (**U**) [25], and MNIST-M (**M-M**) [14]. To perform the pretraining, we utilize the spectral contrastive loss with zero-mean regularization on the PreAct-ResNet18 architecture. The models are pretrained over all the domains for 200 epochs. Subsequently, we train these pretrained models using a linear classifier on the source domains for 10 epochs. The learning rate is set to 0.005, and the weight decay is set to 1e-5. The results of these experiments are reported in Table 2.

For the experiments on the DomainNet dataset, we follow a similar approach. We pretrain the ResNet-50 architecture over four domains of DomainNet for 200 epochs. Then, we train a linear classifier on the source domains for 100 epochs with base size 2048. The learning rate is set to 0.1, and the weight decay is set to 1e-5. The results of these experiments are provided in Table 10.

Table 10: Top-1 accuracy under linear probe evaluation (%) on all individual domain pairs of DomainNet. The results of the spectral contrastive loss with zero mean regularization $\tau > 0$ with better performance than baseline (the original spectral contrastive loss $\tau = 0$) are **highlighted**. The best results are underlined. ERM denotes empirical risk minimization over source domains.

| Source | Clipart | | | Painting | | | Real | | | Sketch | | | Avg |
|---|---|---|---|---|---|---|---|---|---|---|---|---|---|
| Target | Painting | Real | Sketch | Clipart | Real | Sketch | Clipart | Painting | Sketch | Clipart | Painting | Real | |
| ERM | 5.75 | 13.56 | 11.45 | 13.35 | 17.33 | 10.95 | 27.14 | 17.74 | 14.61 | 18.58 | 6.96 | 11.03 | 14.04 |
| Baseline | 14.22 | 21.55 | 15.97 | 14.85 | 25.62 | 13.92 | 21.46 | 22.74 | 15.03 | 20.58 | 14.81 | 19.15 | 18.33 |
| $\tau = 0.1$ | **14.79** | **22.69** | **16.44** | **15.72** | **27.05** | **14.20** | **22.23** | **23.04** | **15.22** | **21.30** | **16.26** | **21.03** | **19.16** |
| $\tau = 0.2$ | **14.57** | **23.10** | **16.72** | **15.90** | **27.69** | **14.22** | **22.43** | **23.41** | **15.29** | **21.45** | **16.62** | **21.29** | **19.39** |
| $\tau = 0.5$ | 14.16 | **22.37** | **16.49** | **15.32** | **26.99** | 13.44 | **22.67** | **22.92** | 14.72 | 20.34 | **15.84** | **20.62** | **18.82** |
| $\tau = 1.0$ | 11.33 | 19.20 | 13.82 | 12.54 | 23.21 | 11.33 | 19.87 | 19.71 | 12.12 | 16.88 | 13.30 | 16.51 | 15.81 |

**Experimental Results.** As depicted in Table 10, the results demonstrate that unsupervised domain adaptation (UDA) utilizing spectral contrastive pretraining yields superior performance compared to the standard empirical risk minimization (ERM) approach. This indicates that spectral contrastive pretraining effectively decouples class information across different domains, leading to improved adaptation. Furthermore, the results indicate that utilizing zero-mean regularization with different values of $\tau$ (0.1, 0.2, and 0.5) outperforms the baselines. This suggests that zero-mean regularization enables the learning of more discriminative representations, resulting in enhanced performance in the unsupervised domain adaptation task.

Table 11: Top-1 validation accuracy (mean ±std, %) of supervised spectral contrastive loss with zero-mean regularization in Equation 3.8 on benchmark datasets with symmetric label noise. The results of the spectral contrastive loss with zero mean regularization $\tau > 0$ with better performance than the original spectral contrastive loss $\tau = 0$ are **highlighted**. The best results are underlined.

| Datasets | Methods | Symmetric Noise Rate | | | |
|---|---|---|---|---|---|
| | | 0.2 | 0.4 | 0.6 | 0.8 |
| CIFAR-10 | CE | $82.88 \pm 0.04$ | $68.99 \pm 0.15$ | $52.27 \pm 0.20$ | $51.62 \pm 0.08$ |
| | Focal | $60.91 \pm 0.07$ | $46.73 \pm 0.13$ | $29.47 \pm 0.04$ | $13.54 \pm 0.03$ |
| | GCE | $91.35 \pm 0.04$ | $89.14 \pm 0.01$ | $80.50 \pm 0.07$ | $52.25 \pm 0.03$ |
| | Vanilla SpeCL | $90.80 \pm 0.01$ | $88.29 \pm 0.13$ | $85.87 \pm 0.07$ | $75.51 \pm 0.27$ |
| | $\tau = 0.1$ | $\mathbf{91.18 \pm 0.07}$ | $\mathbf{88.99 \pm 0.07}$ | $\mathbf{86.66 \pm 0.04}$ | $\underline{\mathbf{80.29 \pm 0.39}}$ |
| | $\tau = 0.2$ | $\underline{\mathbf{91.57 \pm 0.03}}$ | $\mathbf{89.13 \pm 0.12}$ | $\mathbf{86.81 \pm 0.03}$ | $\mathbf{80.06 \pm 0.39}$ |
| | $\tau = 0.5$ | $\mathbf{91.31 \pm 0.06}$ | $\underline{\mathbf{89.49 \pm 0.02}}$ | $\mathbf{86.59 \pm 0.09}$ | $\mathbf{78.51 \pm 0.25}$ |
| | $\tau = 1.0$ | $\mathbf{91.42 \pm 0.07}$ | $\mathbf{89.26 \pm 0.05}$ | $\underline{\mathbf{86.98 \pm 0.07}}$ | $\mathbf{76.63 \pm 0.30}$ |
| | $\tau = 5.0$ | $90.26 \pm 0.02$ | $87.23 \pm 0.10$ | $84.36 \pm 0.02$ | $\mathbf{76.32 \pm 0.07}$ |
| | $\tau = 10.0$ | $84.51 \pm 0.05$ | $80.52 \pm 0.13$ | $77.43 \pm 0.20$ | $68.99 \pm 0.55$ |
| CIFAR-100 | CE | $60.94 \pm 0.06$ | $46.07 \pm 0.01$ | $30.33 \pm 0.02$ | $14.41 \pm 0.06$ |
| | Focal | $60.91 \pm 0.07$ | $46.73 \pm 0.13$ | $29.47 \pm 0.04$ | $13.54 \pm 0.03$ |
| | GCE | $68.91 \pm 0.03$ | $64.87 \pm 0.07$ | $56.04 \pm 0.04$ | $9.13 \pm 0.00$ |
| | Vanilla SpeCL | $67.87 \pm 0.05$ | $62.97 \pm 0.11$ | $57.51 \pm 0.02$ | $51.10 \pm 0.08$ |
| | $\tau = 0.1$ | $\mathbf{68.63 \pm 0.09}$ | $\mathbf{64.46 \pm 0.04}$ | $\underline{\mathbf{59.41 \pm 0.06}}$ | $\mathbf{52.32 \pm 0.09}$ |
| | $\tau = 0.2$ | $\underline{\mathbf{69.60 \pm 0.07}}$ | $\mathbf{65.04 \pm 0.07}$ | $\underline{\mathbf{59.46 \pm 0.04}}$ | $\underline{\mathbf{53.90 \pm 0.06}}$ |
| | $\tau = 0.5$ | $\mathbf{69.57 \pm 0.06}$ | $\mathbf{65.05 \pm 0.06}$ | $\mathbf{59.17 \pm 0.09}$ | $\mathbf{52.80 \pm 0.08}$ |
| | $\tau = 1.0$ | $\mathbf{69.50 \pm 0.15}$ | $\underline{\mathbf{65.37 \pm 0.07}}$ | $\mathbf{59.41 \pm 0.01}$ | $\mathbf{52.60 \pm 0.07}$ |
| | $\tau = 5.0$ | $66.56 \pm 0.03$ | $60.78 \pm 0.07$ | $53.96 \pm 0.09$ | $47.28 \pm 0.06$ |
| | $\tau = 10.0$ | $56.01 \pm 0.08$ | $48.72 \pm 0.11$ | $43.96 \pm 0.09$ | $38.09 \pm 0.12$ |

## B.5 EXPERIMENTS ON LEARNING WITH NOISY LABELS

For the experiments on learning with noisy labels, we propose to employ a mixed spectral contrastive loss to mitigate label noise, *i.e.*, using $\mathcal{L}^{ssl} + 0.5\mathcal{L}^{sup}_{noise}$. The self-supervised part only aligns positive pairs (two random augmentations from the same sample), while the supervised part utilizes positive pairs according to noisy labels.

**Noise generation.** The noisy labels are generated following standard approaches in works [40, 60]. For symmetric label noise, we corrupt the training labels by flipping labels in each class randomly to incorrect labels to incorrect labels in other classes with probability $\eta \in \{0.2, 0.4, 0.6, 0.8\}$.

**Experimental Results.** Table 11 presents the Top-1 validation accuracy in percentage (mean±std of the last five epochs) for various levels of symmetric label noise. As can be seen, the presence of $\tau \leq 1$ consistently improves upon the baseline ($\tau = 0$) across all scenarios.

## B.6 EXPERIMENTS ON "CLASS-MEAN FEATURES AS CLASSIFIER"

To demonstrate the feasibility of employing class-mean features as the classifier following pre-training the representation model $f$ with zero-regularized spectral contrastive losses in the self-supervised or supervised scenarios, we derive the class means $\{\hat{h}_c\}_{c=1}^r$ directly from the average features of each class in training set. We subsequently evaluate its performance using the class prediction rule $\arg\max_c \hat{h}^\top f(x)$ for each test sample $x$. As observed in Table 12, our method, termed as "class-mean Features as classifiers", achieves comparable performance to the conventional linear probing method which trains a linear classifier CE over tens of epochs.

Table 12: Top-1 liner probing (%) and Top-1 validation accuracy (%) of class-mean classifier on self-supervised learning and supervised learning with zero-regularized spectral contrastive loss, respectively. The results with $\tau > 0$ obtaining better performance than the original spectral contrastive loss $\tau = 0$ are **highlighted**. The best results are underlined.

| Method | Self-Supervised Learning | | | Supervised Learning | | |
|--------|----------|-----------|------|----------|-----------|------|
| | **CIFAR-10** | **CIFAR-100** | **SVHN** | **CIFAR-10** | **CIFAR-100** | **SVHN** |
| SpeCL | 84.45 | 46.78 | 88.43 | 92.71 | 57.01 | 96.00 |
| $\tau = 0.1$ | **85.17** | **48.14** | **89.99** | **94.07** | **63.55** | **96.26** |
| $\tau = 0.2$ | **84.90** | **49.54** | **89.81** | **94.38** | **62.23** | **96.30** |
| $\tau = 0.5$ | **85.20** | **49.71** | **89.67** | **94.31** | **63.71** | **96.21** |
| $\tau = 1.0$ | **85.06** | **49.62** | **89.98** | **94.23** | **63.81** | **96.09** |

## C  PROOF OF THEOREMS, LEMMAS AND PROPOSITIONS

### C.1  PROOF OF THEOREM A.1

**Theorem A.1** *Let $\mathcal{F}$ be a class of functions of $\mathcal{X} \to \mathbb{R}^d$ and let $\hat{f} = \arg\min_{f \in \mathcal{F}} \mathcal{L}(f; \tau)$ be the minimizer of the spectral contrastive loss. Assume that:*

1. *($\epsilon$-separability). The probability of a positive pair belonging to two different sets is less than $\epsilon$, that is, $\Pr_{(x,x^+) \sim p_{pos}}(\mathrm{id}_x \neq \mathrm{id}_{x^+}) \leq \epsilon$;*

2. *(Alignment) The downstream label $y(x)$ is a constant on each $\mathcal{S}_i$ for $i \in [m]$;*

3. *($\mathcal{F}$-implementable inner-cluster connection larger than $\delta$). For any $f \in \mathcal{F}$ and any linear head $w \in \mathbb{R}^d$, let function $g(x) = w^\top f(x)$. For any $i \in [m]$, we have $Q_{\mathcal{S}_i}(g) := \frac{\mathbb{E}_{(x,x^+) \sim p_{pos}^{\mathcal{S}_i}}[(g(x)-g(x^+))^2]}{\mathbb{E}_{x \sim p_{data}^{\mathcal{S}_i}, x' \sim p_{data}^{\mathcal{S}_i}}[(g(x)-g(x'))^2]} \geq \delta$;*

4. *(Implementability). There exists a function $f \in \mathcal{F}$ such that $f(x) = v_{id_x}$ for all $x \sim p_{data}$, where $\{v_1, v_2, ..., v_m\}$ is a set of different vectors that achieve the global minimum of $\|\sum_{i=1}^m p_i v_i v_i^\top - \mathrm{I}\|_F^2 + 2\tau \cdot \|\sum_{i=1}^m p_i v_i\|_2^2$, and $p_i = \Pr_{x \sim p_{data}}(x \in \mathcal{S}_i)$.*

*Let $p_{\min} = \min_{i \in [m]} \Pr_{x \sim p_{data}}(x \in \mathcal{S}_i)$ and $\Delta = \max_{i,j} \|v_i - v_j\|_2^2$. For $m > d$, if $\epsilon\Delta < 1$, then there exists a linear head $W \in \mathbb{R}^{d \times d}$ which achieves the following downstream error*

$$\mathbb{E}_{x \sim p_{data}}\left[\left\|W\hat{f}(x) - v_{y(x)}\right\|_2^2\right] \leq \frac{\epsilon\Delta(1 + \sqrt{\epsilon\Delta})p_{\min}}{2\delta(p_{\min} - \epsilon)}. \tag{C.1}$$

*Proof.* Let $f^*$ be the function $f^*(x) = v_{\mathrm{id}_x}$. According to Lemma C.2, if $m > d$, the minimum of $\|\sum_{i=1}^m p_i v_i v_i^\top - \mathrm{I}\|_F^2 + 2\tau \cdot \|\sum_{i=1}^m p_i v_i\|_2^2$ is 0, which is obtained when $\sum_{i=1}^m p_i v_i v_i^\top = \mathrm{I}$ and $\sum_{i=1}^m p_i v_i = 0$, so we know that $\mathcal{R}_1(f^*) = \mathcal{R}_2(f^*) = 0$. For the invariance term with respect to $f^*$, we have

$$\mathcal{R}_0(f^*) = \mathbb{E}_{x,x^+}\left[\|f^*(x) - f^*(x^+)\|_2^2\right] \leq \max_{i,j} \|v_i - v_j\|_2^2 \Pr_{(x,x^+) \sim p_{pos}}(\mathrm{id}_x \neq \mathrm{id}_{x^+}) \leq \epsilon\Delta. \tag{C.2}$$

Define matrix $M = \mathbb{E}_{x \sim p_{data}}[\hat{f}(x)\hat{f}(x)^\top]$, since $\hat{f} = \arg\min_{f \in \mathcal{F}} \mathcal{L}(f; \tau)$ is the minimizer of contrastive loss within the functional class, we have $\mathcal{R}_0(\hat{f}) + \mathcal{R}_1(\hat{f}) + 2\tau\mathcal{R}_2(\hat{f}) \leq \mathcal{R}_0(f^*)$, and further obtain

$$\|M - \mathrm{I}\|_F^2 = \mathcal{R}_1(\hat{f}) \leq \mathcal{R}_0(f^*) \leq \epsilon\Delta < 1. \tag{C.3}$$

Thus, $M$ is a full rank matrix, and we can define function $\tilde{f}(x) = M^{-1/2}\hat{f}(x)$ and have that

$$\mathbb{E}_{x \sim p_{data}}[\tilde{f}(x)\tilde{f}(x)^\top] = \mathbb{E}_{x \sim p_{data}}[M^{-1/2}\hat{f}(x)\hat{f}(x)^\top M^{-1/2}] = \mathrm{I}. \tag{C.4}$$

Let $Q = \mathbb{E}_{x \sim p_{data}}[\tilde{f}(x)f^*(x)^\top]$ and $\pi_f(x) = \tilde{f}(x) - Qf^*(x)$. Since $\mathcal{R}_1(f^*) = 0$, we know that

$$\mathbb{E}_{x \sim p_{data}}[\pi_f(x)f^*(x)^\top] = \mathbb{E}_{x \sim p_{data}}[\tilde{f}(x)f^*(x)^\top] - Q\mathbb{E}_{x \sim p_{data}}[f^*(x)f^*(x)^\top] = 0. \tag{C.5}$$

Using the first and third assumptions in Theorem 3.1, we have

$$\mathbb{E}_{(x,x^+)\sim p_{\text{pos}}}\left[\left\|\pi_f(x)-\pi_f(x^+)\right\|_2^2\right]$$

$$\geq \sum_{i\in[m]}(p_i-\epsilon)\cdot\mathbb{E}_{(x,x^+)\sim p_{pos_i}}\left[\left\|\pi_f(x)-\pi_f(x^+)\right\|_2^2\right]$$

$$\geq\delta\cdot\sum_{i\in[m]}(p_i-\epsilon)\cdot\mathbb{E}_{x,x'\sim p_{data_i}}\left[\left\|\pi_f(x)-\pi_f(x')\right\|_2^2\right]$$

$$=\delta\cdot\sum_{i\in[m]}\left(1-\frac{\epsilon}{p_i}\right)p_i\mathbb{E}_{x,x'\sim p_{data_i}}\left[\left\|\pi_f(x)-\pi_f(x')\right\|_2^2\right] \tag{C.6}$$

$$\geq\delta\cdot\left(1-\frac{\epsilon}{p_{\min}}\right)\sum_{i\in[m]}p_i\mathbb{E}_{x,x'\sim p_{data_i}}\left[\left\|\pi_f(x)-\pi_f(x')\right\|_2^2\right]$$

$$=2\delta\cdot\left(1-\frac{\epsilon}{p_{\min}}\right)\cdot\mathbb{E}_{x\sim p_{\text{data}}}\left[\left\|\pi_f(x)\right\|_2^2\right].$$

On the other hand, we have

$$\mathbb{E}_{(x,x^+)\sim p_{\text{pos}}}\left[\left\|\pi_f(x)-\pi_f(x^+)\right\|\right]$$

$$\leq\mathbb{E}_{(x,x^+)\sim p_{\text{pos}}}\left[\left\|\tilde{f}(x)-\tilde{f}(x^+)\right\|\right]$$

$$\leq\|M^{-1}\|_{spec}\cdot\mathbb{E}_{(x,x^+)\sim p_{\text{pos}}}\left[\left\|\hat{f}(x)-\hat{f}(x^+)\right\|\right] \tag{C.7}$$

$$\leq(1+\sqrt{\epsilon\Delta})\cdot\epsilon\Delta.$$

Combining Equation C.6 and C.7 we have

$$\mathbb{E}_{x\sim p_{\text{data}}}\left[\|\pi_f(x)\|_2^2\right]\leq\frac{\epsilon\Delta(1+\sqrt{\epsilon\Delta})p_{\min}}{2\delta(p_{\min}-\epsilon)}. \tag{C.8}$$

By Lemma C.1, we know that there exists a matrix $U\in\mathbb{R}^{d\times d}$ such that

$$\mathbb{E}_{x\sim p_{\text{data}}}[\|f^*(x)-UM^{-1/2}\hat{f}(x)\|_2^2]\leq\frac{\epsilon\Delta(1+\sqrt{\epsilon\Delta})p_{\min}}{2\delta(p_{\min}-\epsilon)}. \tag{C.9}$$

Thus, if we define matrix $W=UM^{-1/2}$, then we have

$$\mathbb{E}_{x\sim p_{\text{data}}}\left[\|v_{\text{id}_x}-W\hat{f}(x)\|_2^2\right]\leq\frac{\epsilon\Delta(1+\sqrt{\epsilon\Delta})p_{\min}}{2\delta(p_{\min}-\epsilon)}, \tag{C.10}$$

which finishes the proof when $d<m$. □

**Lemma C.1.** *Suppose $f:\mathcal{X}\to\mathbb{R}^d$ and $g:\mathcal{X}\to\mathbb{R}^d$ are two functions defined on $\mathcal{X}$ such that*

$$\mathbb{E}_{x\sim p_{\text{data}}}[f(x)f(x)^\top]=\mathbb{E}_{x\sim p_{\text{data}}}[g(x)g(x)^\top]=\mathrm{I}. \tag{C.11}$$

*Define the projection of $f$ onto $g$'s orthogonal subspace as*

$$\pi_f(x)=f(x)-\mathbb{E}_{x'\sim p_{\text{data}}}[f(x')g(x')^\top]g(x). \tag{C.12}$$

*Then, there exist matrix $U\in\mathbb{R}^{d\times d}$ such that*

$$\mathbb{E}_{x\sim p_{\text{data}}}[\|g(x)-Uf(x)\|_2^2]=\mathbb{E}_{x\sim p_{\text{data}}}[\|\pi_f(x)\|_2^2]. \tag{C.13}$$

*Proof.* Please see the proof in HaoChen and Ma [18, Lemma B.1]. □

**Lemma C.2.** *For any given weights $p_1,...,p_m>0$ satisfying $\sum_{i=1}^m p_i=1$ and $\tau\in[0,1]$, the global minimum of $\|\sum_{i=1}^m p_i v_i v_i^\top-\mathrm{I}_d\|_F^2+2\tau\cdot\|\sum_{i=1}^m p_i v_i\|_2^2$ (where $d$ is the feature dimensionality) is*

*(a) 0 when $m>d$, which is obtained if and only if $VP[V;1_m]^\top=[\mathrm{I}_d;0]$;*

*(b) $d - m + 2\tau - \tau^2$ when $m \le d$, which is obtained if and only if $V^\top V = P^{-1} - \tau 1_m 1_m^\top$,*

*where $V = [v_1, ..., v_m]$ and $P = \mathrm{diag}\,(p_1, p_2, ..., p_m)$.*

*Proof.* Obviously, we have $\| \sum_{i=1}^m p_i v_i v_i^\top - \mathrm{I}_d \|_F^2 + 2\tau \cdot \| \sum_{i=1}^m p_i v_i \|_2^2 \ge 0$, where the equality holds if and only if $\sum_{i=1}^m p_i v_i v_i^\top = \mathrm{I}_d$ and $\tau \cdot \sum_{i=1}^m p_i v_i = 0$. However, if $m \le d$, it is almost impossible to achieve this condition since there are fewer unknowns than equations. Thus, we will analyze the global minimizer while considering the relation between $m$ and $d$.

If $m > d$, there exist different vectors that satisfy $\sum_{i=1}^m p_i v_i v_i^\top = \mathrm{I}_d$ and $\sum_{i=1}^m p_i v_i = 0$, that is, $V P [V^\top; 1_m^\top] = [\mathrm{I}_d; 0]$.

If $m \le d$, we have

$$
\left\| \sum_{i \in [m]} p_i v_i v_i^\top - \mathrm{I}_d \right\|_F^2 + 2\tau \left\| \sum_{i \in [m]} p_i v_i \right\|_2^2
$$

$$
= \mathrm{Tr} \left( \left( \sum_{i \in [m]} p_i v_i v_i^\top - \mathrm{I}_d \right)^\top \left( \sum_{i \in [m]} p_i v_i v_i^\top - \mathrm{I}_d \right) \right) + 2\tau \left\| \sum_{i \in [m]} p_i v_i \right\|_2^2
$$

$$
= \mathrm{Tr} \left( \sum_{i \in [m]} \sum_{j \in [m]} p_i p_j v_i v_i^\top v_j v_j^\top - 2 \sum_{i \in [m]} p_i v_i v_i^\top + \mathrm{I}_d \right) + 2\tau \sum_{i \in [m]} \sum_{j \in [m]} p_i p_j v_i^\top v_j
$$

$$
= \sum_{i \in [m]} \sum_{j \in [m]} p_i p_j (v_i^\top v_j)^2 - 2 \sum_{i \in [m]} p_i v_i^\top v_i + d + 2\tau \sum_{i \in [m]} \sum_{j \in [m]} p_i p_j v_i^\top v_j \tag{C.14}
$$

$$
= \sum_{i \in [m]} \sum_{j \in [m]} p_i p_j (v_i^\top v_j + \tau)^2 - \tau^2 - 2 \sum_{i \in [m]} p_i v_i^\top v_i + d
$$

$$
= d + \sum_{i \ne j} p_i p_j (v_i^\top v_j + \tau)^2 - \tau^2 + \sum_{i \in [m]} p_i^2 (\|v_i\|_2^2 + \tau)^2 - 2 \sum_{i \in [m]} p_i \|v_i\|_2^2
$$

$$
= d - \tau^2 + \sum_{i \ne j} p_i p_j (v_i^\top v_j + \tau)^2 + \sum_{i \in [m]} \left( p_i \|v_i\|_2^2 + \tau p_i - 1 \right)^2 + 2\tau - m
$$

$$
= d - m + 2\tau - \tau^2 + \sum_{i \ne j} p_i p_j (v_i^\top v_j + \tau)^2 + \sum_{i \in [m]} \left( p_i \|v_i\|_2^2 + \tau p_i - 1 \right)^2 .
$$

Thus, the global minimum of $\| \sum_{i=1}^m p_i v_i v_i^\top - \mathrm{I}_d \|_F^2 + 2\tau \cdot \| \sum_{i=1}^m p_i v_i \|_2^2$ is $d - m + 2\tau - \tau^2$, which is obtained at

$$
v_i^\top v_j = -\tau, \ \forall i, j \in [m] \text{ and } i \ne j, \quad p_i \|v_i\|_2^2 + \tau p_i - 1 = 0, \ \forall i \in [m], \tag{C.15}
$$

that is, $V^\top V = P^{-1} - \tau 1_m 1_m^\top$, and further

$$
\| \sum_{i \in [m]} p_i v_i v_i^\top - \mathrm{I}_d \|_F^2 = d - m + \tau^2, \text{ and } \| \sum_{i \in [m]} p_i v_i \|_2^2 = 1 - \tau, \tag{C.16}
$$

and this equation requires that $\tau \le \inf_{z \in \mathbb{R}^m} \frac{z^\top P z}{z^\top 1_m 1_m^\top z} = 1$. $\qquad \square$

## C.2    PROOF OF THEOREM 3.1 AND PROPOSITION 3.2

**Theorem 3.1** *Let $\zeta > 0$ and $\epsilon \in (0, \frac{1}{2})$ be arbitrary constants. In the above stochastic block model, assume $\rho > \max\{\alpha, \beta\}$, $\gamma < \min\{\alpha, \beta\}$ and $\tau < \frac{\tilde{\lambda}_1(0) - \tilde{\lambda}_d(0)}{\tilde{\lambda}_1(0)}$. Then, there exists $\tilde{\xi} \in [1 - \epsilon, 1]$, such that for any $n \ge \Omega \left( \frac{rp}{\min\{\alpha - \gamma, \beta - \gamma\}^2} \right)$ and regularization strength $\eta \in \left( 0, \frac{(\alpha - \gamma)\epsilon}{2r\rho} \right]$, with a high probability $1 - n^{-\zeta}$, we have*

$$
\left\| \mathrm{pred}_\mathcal{T} - \widetilde{\mathrm{pred}}_\mathcal{T} \right\|_F \le O \left( \frac{\tilde{\lambda}_1(\tau)}{\eta^2 (\tilde{\lambda}_d(\tau) - \tilde{\lambda}_{d+1}(\tau))} \right) \cdot \mathrm{poly}(r, p), \tag{C.17}
$$

where $\widetilde{\mathrm{pred}}_{\mathcal{T}} \triangleq \mathbb{E}[\mathrm{pred}_{\mathcal{T}}]$ *is the expectation of the prediction matrix when achieving the minimum of the loss in Eq. 7 with* $\tau \in [0,1]$, $\tilde{\lambda}_1(\tau), \tilde{\lambda}_d(\tau)$ *and* $\tilde{\lambda}_{d+1}(\tau)$ *are the 1-st, d-th and $d+1$-th eigenvalues of* $\tilde{A} \triangleq \mathbb{E}\left[A - \frac{\tau|\tilde{E}|}{N^2}11^\top\right]$, *respectively.*

*Furthermore, the target error can be bounded by*

$$\mathbb{P}_{x \sim \mathcal{T}}(\mathrm{pred}(x) \neq y_x) \leq O\left(\frac{(\tilde{\lambda}_1(\tau))^2}{\eta^4(\tilde{\lambda}_d(\tau) - \tilde{\lambda}_{d+1}(\tau))^2 \cdot n}\right) \cdot \mathrm{poly}(r,p), \tag{C.18}$$

*where* $\mathrm{poly}(r,p)$ *denotes a polynomial function of r and p.*

*Proof.* Let $A_d \in \mathbb{R}^{N \times N}$ be the rank-$d$ approximation of the adjacency matrix $A$, which contains the top-$d$ components of $A$'s SVD decomposition. We use $A_{d,(\mathcal{T},\mathcal{S})}$ to denote the matrix by restricting $A_d$ to the rows corresponding to the source and the columns corresponding to the target. We use $A_{d,(\mathcal{S},\mathcal{S})}$ to denote the matrix by restricting $A_d$ to the rows and columns corresponding to the source.

Let $Y \in \mathbb{R}^{N \times r}$ be the label matrix where on the $x$-th row it contains the target of the label $\vec{y}_x$. Let $Y_{\mathcal{S}} \in \mathbb{R}^{|\mathcal{S}| \times r}$ and $T_{\mathcal{T}} \in \mathbb{R}^{|\mathcal{T}| \times r}$ be the matrices by restricting $Y$ to the source and target domains, respectively.

We can rewrite the spectral contrastive loss as

$$-2\sum_{x,x'} \frac{A_{xx'}}{|E|} f(x)^\top f(x') + \sum_{x,x'} \frac{1}{N^2}\left(f(x)^\top f(x') + \tau\right)^2$$

$$= \left\|\frac{N}{|E|} \cdot \left(A - \frac{\tau|E|}{N^2}11^\top\right) - \left(\frac{1}{\sqrt{N}} \cdot F\right)\left(\frac{1}{\sqrt{N}} \cdot F\right)^\top\right\|_F^2 + const. \tag{C.19}$$

Let $A' = A - \frac{\tau|E|}{N^2}11^\top$, $\tilde{A} = \mathbb{E}[A]$, $|\tilde{E}| = 1_N^\top \tilde{A} 1_N$ be the expectation of number of edges in the graph $A$, $\tilde{A}' = \mathbb{E}[A'] = \mathbb{E}\left[A - \frac{\tau|\tilde{E}|}{N^2}11^\top\right]$, and $\tilde{\lambda}_i(\tau)$ be the $i$-th eigenvalue of $\tilde{A}'$. By Lemma C.3, we have that the prediction on the target domain is

$$\mathrm{pred}_{\mathcal{T}} = A'_{d,(\mathcal{T},\mathcal{S})}\left(A'_{d,(\mathcal{S},\mathcal{S})} + \frac{|E|}{N^2} \cdot \eta \cdot |\mathcal{S}| \cdot \mathrm{I}\right)^\dagger Y_{\mathcal{S}}. \tag{C.20}$$

We define the ideal prediction as

$$\widetilde{\mathrm{pred}}_{\mathcal{T}} = \tilde{A}'_{d,(\mathcal{T},\mathcal{S})}\left(\tilde{A}'_{d,(\mathcal{S},\mathcal{S})} + \frac{|\tilde{E}|}{N^2} \cdot \eta \cdot |\mathcal{S}| \cdot \mathrm{I}\right)^\dagger Y_{\mathcal{S}}. \tag{C.21}$$

We will bound the difference between $\mathrm{pred}_{\mathcal{T}}$ and $\widetilde{\mathrm{pred}}_{\mathcal{T}}$. For every class $c \in [r]$, define the following error vector

$$v^c \triangleq A'_{d,(\mathcal{T},\mathcal{S})}\left(A'_{d,(\mathcal{S},\mathcal{S})} + \frac{|E|}{N^2} \cdot \eta \cdot |\mathcal{S}| \cdot I\right)^\dagger Y_{\mathcal{S}}^c - \tilde{A}'_{d,(\mathcal{T},\mathcal{S})}\left(\tilde{A}'_{d,(\mathcal{S},\mathcal{S})} + \frac{|E|}{N^2} \cdot \eta \cdot |\mathcal{S}| \cdot I\right)^\dagger Y_{\mathcal{S}}^c,$$

where $Y_{\mathcal{S}}^c$ is the $c$-th column of $Y_{\mathcal{S}}$.

Using the perturbation bound for the pseudoinverse matrix [47], we have

$$\|v^c\| \leq \|A'_{d,(\mathcal{T},\mathcal{S})} - \tilde{A}'_{d,(\mathcal{T},\mathcal{S})}\| \cdot \left\|\left(A'_{d,(\mathcal{S},\mathcal{S})} + \frac{|E|}{N^2} \cdot \eta \cdot |\mathcal{S}| \cdot I\right)^\dagger\right\| \cdot \|Y_{\mathcal{S}}^c\|$$

$$+ \|A'_{d,(\mathcal{T},\mathcal{S})}\| \cdot \left\|\left(A'_{d,(\mathcal{S},\mathcal{S})} + \frac{|E|}{N^2} \cdot \eta \cdot |\mathcal{S}| \cdot I\right)^\dagger - \left(\tilde{A}'_{d,(\mathcal{S},\mathcal{S})} + \frac{|E|}{N^2} \cdot \eta \cdot |\mathcal{S}| \cdot I\right)^\dagger\right\| \cdot \|Y_{\mathcal{S}}^c\|$$

$$\leq \left(\frac{n^2}{\eta|E| \cdot |\mathcal{S}|} + \frac{1+\sqrt{5}}{2} \cdot \left(\frac{n^2}{\eta|E| \cdot |\mathcal{S}|}\right)^2 \cdot \|A'_d\|\right) \cdot \|Y_{\mathcal{S}}^c\| \cdot \|A'_d - \tilde{A}'_d\|, \tag{C.22}$$

where the first inequality is induced by the triangle inequality[5], the second inequality in Eq. C.22 is based the facts that $\|A'_{d,(\mathcal{T},\mathcal{S})}\| \leq \|A'_d\|$, $\|A'_{d,(\mathcal{T},\mathcal{S})} - \tilde{A}_{d,(\mathcal{T},\mathcal{S})}\| \leq \|A'_d - \tilde{A}'_d\|$, $\|(A'_{d,(\mathcal{S},\mathcal{S})} + \frac{\eta|E|\cdot|\mathcal{S}|}{N^2} \cdot I)^\dagger\| \leq \frac{N^2}{\eta|E|\cdot|\mathcal{S}|}$, and the upper bound for the difference between two pseudoinverse matrix $\|B^\dagger - A^\dagger\| \leq \frac{1+\sqrt{5}}{2} \max\{\|A^\dagger\|, \|B^\dagger\|\} \cdot \|B - A\|$ [47, Theorem 3.3].

By Lemma C.6, there exists constant $C = C(\zeta)$ such that $\|A' - \tilde{A}'\| \leq C\sqrt{N}$ with probability at least $1 - N^{-2\zeta}$ for any $N > \Omega(1/\rho)$. From now on, we assume this high probability event happens.

Follow some notations in the proof of Lemma C.4, we have $\tilde{\lambda}_d(\tau) = \min\{\lambda_\diamond, \lambda_\dagger\}$ and $\tilde{\lambda}_{d+1}(\tau) = \lambda_z$, which is based on the facts that the eigenvalue corresponding to the eigenvector $\overline{1}_p \otimes \overline{1}_r \otimes \overline{1}_n$ of $\tilde{A}'$ satisfies

$$n\rho + n(r-1)\beta + n(p-1)\alpha + n(p-1)(r-1)\gamma - \frac{\tau|\tilde{E}|}{N} = (1-\tau)\tilde{\lambda}_1(0) > \tilde{\lambda}_d(0), \quad \text{(C.23)}$$

when we have $\tilde{\lambda}_1(0) = n\rho + n(r-1)\beta + n(p-1)\alpha + n(p-1)(r-1)\gamma$, $|\tilde{E}| = 1_N^\top \tilde{A} 1_N = (\beta-\gamma)pr^2n^2 + (\alpha-\gamma)p^2rn^2 + (\rho-\beta-\alpha+\gamma)prn^2 + \gamma p^2r^2n^2 = N\tilde{\lambda}_1(0)$, and $\tau < \frac{\tilde{\lambda}_1(0)-\tilde{\lambda}_d(0)}{\tilde{\lambda}_1(0)}$.

Let $\Delta \triangleq \frac{\tilde{\lambda}_d(\tau)-\tilde{\lambda}_{d+1}(\tau)}{n} = \min\{r(\beta-\gamma), p(\alpha-\gamma)\}$. If our choice of $N$ further satisfies $N \geq \left(\frac{2rmC}{\Delta}\right)^2$, we have $\|A' - \tilde{A}'\| \leq \frac{1}{2}(\tilde{\lambda}_d(\tau) - \tilde{\lambda}_{d+1}(\tau))$, so from Lemma C.5, we have

$$\|A'_d - \tilde{A}'_d\| \leq O\left(\frac{\tilde{\lambda}_1(\tau)}{\tilde{\lambda}_d(\tau) - \tilde{\lambda}_{d+1}(\tau)} \cdot \|A' - \tilde{A}'\|\right) \leq O\left(\frac{\tilde{\lambda}_1(\tau)}{\tilde{\lambda}_d(\tau) - \tilde{\lambda}_{d+1}(\tau)} \cdot \sqrt{N}\right). \quad \text{(C.24)}$$

By Hoeffding's inequality, with probability at least $1 - 2e^{-2N^2}$ we assume $||E| - |\tilde{E}|| \leq N$. From now on, we assume this high-probability event happens. The total failure probability so far is $N^{-2\zeta} + 2e^{-2N^2} \leq N^{-\zeta}$. By the definition of graph, we have $|\tilde{E}| \geq \frac{\rho N^2}{rp}$. If our choice of $N$ further satisfies $N \geq \frac{2rp}{\rho}$, we have $|E| \geq \frac{\rho N^2}{2rp}$, thus

$$\frac{|E| \cdot |\mathcal{S}|}{N^2} \geq \frac{\rho N}{2rp^2}. \quad \text{(C.25)}$$

Substituting Equation C.24 and Equation C.25 into Equation C.22 gives:

$$\|v^c\| \leq O\left(\frac{\tilde{\lambda}_1(\tau)}{\eta^2(\tilde{\lambda}_d(\tau) - \tilde{\lambda}_{d+1}(\tau))\sqrt{N}}\right) \cdot \text{poly}(r,p) \cdot \|Y_{\mathcal{S}}^c\|. \quad \text{(C.26)}$$

Summing over all classes $c$ and noticing that $\|Y_{\mathcal{S}}^c\| \leq \sqrt{N}$ leads to

$$\|v\|_F = \sqrt{\sum_{c=1}^r \|v^c\|^2} \leq \sum_{c=1}^r \|v^c\| \leq O\left(\frac{\tilde{\lambda}_1(\tau)}{\eta^2(\tilde{\lambda}_d(\tau) - \tilde{\lambda}_{d+1}(\tau))}\right) \cdot \text{poly}(r,p). \quad \text{(C.27)}$$

Moreover, let $\xi = \frac{|E|}{N^2} \cdot \eta \cdot |\mathcal{S}|$ and $\tilde{\xi} = \frac{|\tilde{E}|}{N^2} \cdot \eta \cdot |\mathcal{S}|$, we have

$$\left|\frac{\lambda_\dagger}{\lambda_\dagger + p\xi} - \frac{\lambda_\dagger}{\lambda_\dagger + p\tilde{\xi}}\right| \leq \frac{p}{\lambda_\dagger} \cdot |\xi - \tilde{\xi}| = \frac{\eta||E| - |\tilde{E}||}{\lambda_\dagger N} \leq \text{poly}(r,p). \quad \text{(C.28)}$$

By Lemma C.4, we have

$$\left\|\tilde{A}'_{d,(\mathcal{T},\mathcal{S})}\left(\tilde{A}'_{d,(\mathcal{S},\mathcal{S})} + \frac{|E|}{N^2}\cdot\eta\cdot|\mathcal{S}|\cdot I\right)^\dagger Y_{\mathcal{S}} - \tilde{A}'_{d,(\mathcal{T},\mathcal{S})}\left(\tilde{A}'_{d,(\mathcal{S},\mathcal{S})} + \frac{|\tilde{E}|}{N^2}\cdot\eta\cdot|\mathcal{S}|\cdot I\right)^\dagger Y_{\mathcal{S}}\right\|_F$$

$$= \left|\frac{\lambda_\dagger}{\lambda_\dagger + p\xi} - \frac{\lambda_\dagger}{\lambda_\dagger + p\tilde{\xi}}\right| \cdot \|Y_{\mathcal{T}}\|_F \leq \frac{p}{\lambda_\dagger} \cdot |\xi - \tilde{\xi}| \cdot \|Y_{\mathcal{T}}\|_F \leq \frac{1}{\sqrt{N}}\text{poly}(r,p). \quad \text{(C.29)}$$

---

[5]That is, $\|AB - CD\| = \|AB - AD + AD - CD\| \leq \|A\|\|B - D\| + \|A - C\|\|D\|$.

Combining Equation C.27 and Equation C.29, we have

$$\left\| \text{pred}_{\mathcal{T}} - \widetilde{\text{pred}}_{\mathcal{T}} \right\|_F \leq O\left( \frac{\tilde{\lambda}_1(\tau)}{\eta^2(\tilde{\lambda}_d(\tau) - \tilde{\lambda}_{d+1}(\tau))} \right) \cdot \text{poly}(r, p), \tag{C.30}$$

Notice that the $x$-th row of $\widetilde{\text{pred}}_{\mathcal{T}}$ is $\frac{\lambda_{\dagger}}{\lambda_{\dagger} + p\tilde{\zeta}} \cdot y_{y_x}$. Since $\lambda_{\dagger} = n(\rho - \beta + (p-1)\alpha - (p-1)\gamma) \geq \frac{1}{2} np(\alpha - \gamma)$, and $\tilde{\xi} = \frac{|\tilde{E}|}{N^2} \cdot \eta \cdot |\mathcal{S}| \leq \eta r n \rho$, we have $\frac{\lambda_{\dagger}}{\lambda_{\dagger} + p\tilde{\zeta}} \geq \frac{1}{1 + \frac{2r\rho\eta}{p(\alpha-\gamma)}} \geq 1 - \epsilon$, where the second inequality follows the assumption on $\eta$.

Since $\text{pred}_{\mathcal{T}}$ is incorrect on the $x$-th row only if its difference from the $x$-th row of $\widetilde{\text{pred}}_{\mathcal{T}}$ has larger norm than $\Omega(1 - \epsilon)$, we know the final total error on the target domains is bounded by $O\left( \frac{(\tilde{\lambda}_1(\tau))^2}{\eta^4(\tilde{\lambda}_d(\tau) - \tilde{\lambda}_{d+1}(\tau))^2 \cdot n} \right) \cdot \text{poly}(r, p)$.

Collecting all the requirements of $N$, this bound holds so long as $N \geq \Omega\left( \left( \frac{rp}{\min\{\alpha - \gamma, \beta - \gamma\}} \right)^2 \right)$, which is equivalent to $n \geq \Omega\left( \frac{rp}{\min\{\alpha - \gamma, \beta - \gamma\}} \right)$ as $N = rpn$. $\qquad\square$

**Proposition 3.2** *In the setting of Theorem 3.1, if $\tau \leq \frac{\tilde{\lambda}_1(0) - \tilde{\lambda}_2(0)}{\tilde{\lambda}_1(0)} = \frac{n \cdot \min\{p\alpha + r(p-1)\gamma, r\beta + p(r-1)\gamma\}}{\tilde{\lambda}_1(0)}$, then, with probability at least $1 - n^{-\zeta}$, we have*

$$\mathbb{P}_{x \sim \mathcal{T}}(\text{pred}(x) \neq y_x) \leq O\left( \frac{(1-\tau)^2(\tilde{\lambda}_1(0))^2}{\eta^4(\tilde{\lambda}_d(0) - \tilde{\lambda}_{d+1}(0))^2 \cdot n} \right) \cdot \text{poly}(r, p), \tag{C.31}$$

*where $\tilde{\lambda}_1(0) = n\rho + n(r-1)\beta + n(p-1)\alpha + n(p-1)(r-1)\gamma$ and $\tilde{\lambda}_d(0) - \tilde{\lambda}_{d+1}(0) = n \cdot \min\{r(\beta - \gamma), p(\alpha - \gamma)\}$.*

*Proof.* According the proof of Theorem 3.1, we know that when $\tau < \frac{\tilde{\lambda}_1(0) - \tilde{\lambda}_2(0)}{\tilde{\lambda}_1(0)}$, the first eigenvalue of $\tilde{A}'$ will be

$$\tilde{\lambda}_1(\tau) = n\rho + n(r-1)\beta + n(p-1)\alpha + n(p-1)(r-1)\gamma - \frac{\tau|\tilde{E}|}{N} = (1-\tau)\tilde{\lambda}_1(0), \tag{C.32}$$

and $\tilde{\lambda}_d(\tau) = \tilde{\lambda}_d(0) = n(\rho - \beta - \alpha + \gamma) + n \cdot \min\{p(\alpha - \gamma), r(\beta - \gamma)\}$, and $\tilde{\lambda}_{d+1}(\tau) = \tilde{\lambda}_{d+1}(0) = n(\rho - \beta - \alpha + \gamma)$. Substituting these results into Theorem 4.1 finishes the proof. $\qquad\square$

**Lemma C.3.** *For $\hat{f}$ that achieves the minimum of spectral contrastive loss $-2\sum_{x,x'} \frac{A_{x,x'}}{C} f(x)^\top f(x') + \sum_{x,x'} \frac{1}{N^2}(f(x)^\top f(x'))^2$ (where $C > 0$ is any constant), let $\hat{b} = \arg\min_{b \in \mathbb{R}^{d \times r}} \sum_{x \in \mathcal{S}} \left( \|b^\top \hat{f}(x) - \vec{y}_x\|_2^2 + \eta\|b\|_F^2 \right)$ be the linear head learned on the source domain $\mathcal{S}$. Let $\text{pred} \in \mathbb{R}^{N \times r}$ be the matrix with $\hat{b}^\top \hat{f}(x)$ as its $x$-th row and $\text{pred}_{\mathcal{T}}$ be the matrix by restricting $\text{pred}$ to the target domain $\mathcal{T}$, then we have*

$$\text{pred}_{\mathcal{T}} = A_{d,(\mathcal{T},\mathcal{S})} \left( A_{d,(\mathcal{S},\mathcal{S})} + \frac{C}{N^2} \cdot \eta \cdot |\mathcal{S}| \cdot I \right)^{\dagger} Y_{\mathcal{S}}, \tag{C.33}$$

*where $(\cdot)^{\dagger}$ is the Moore-Penrose inverse, $|\mathcal{S}|$ is the number of data in the source domain, and $A_d$ is the rank-$d$ approximation of $A$.*

*Proof.* We can rewrite the spectral contrastive loss as

$$-2\sum_{x,x'} \frac{A_{x,x'}}{C} f(x)^\top f(x') + \sum_{x,x'} \frac{1}{N^2}(f(x)^\top f(x'))^2$$
$$= \left\| \frac{N}{C} \cdot A - \left( \frac{1}{\sqrt{N}} \cdot F \right)^\top \left( \frac{1}{\sqrt{N}} \cdot F \right) \right\|_F^2 + const, \tag{C.34}$$

where $F \in \mathbb{R}^{N \times d}$ is the matrix which the $x$-th row contains $f(x)^\top$. According to the Eckart-Young-Mirsky theorem, the minimizer of the above loss function is $F = \frac{N}{\sqrt{C}} S_d$ where $S_d \in \mathbb{R}^{N \times d}$ is a matrix such that $A_k = S_d^\top S_d$.

Let $S_{d,\mathcal{S}} \in \mathbb{R}^{|\mathcal{S}| \times d}$ be the matrix obtained by restricting $S_d$ to the rows corresponding to the source data, and $S_{d,\mathcal{T}}$ be the matrix obtained by restricting $S_d$ to the rows corresponding to the target data. The head learned on the source domain can be expressed as

$$
\begin{aligned}
\hat{b} &= \arg\min_{b \in \mathbb{R}^{d \times r}} \sum_{x \in \mathcal{S}} \left( \|b^\top f(x) - \vec{y}_x\|_2^2 + \eta \|b\|_F^2 \right) \\
&= \frac{\sqrt{C}}{N} \cdot S_{d,\mathcal{S}}^\top \left( S_{d,\mathcal{S}} S_{d,\mathcal{S}}^\top + \frac{C}{N^2} \cdot \eta \cdot |\mathcal{S}| \cdot \mathrm{I} \right)^\dagger Y_\mathcal{S}.
\end{aligned}
\tag{C.35}
$$

Therefore, the prediction on the target domain $\mathrm{pred}_\mathcal{T}$ is

$$
\begin{aligned}
\mathrm{pred}_\mathcal{T} &= F_\mathcal{T} \hat{b} = S_{d,\mathcal{T}} S_{d,\mathcal{S}}^\top \left( S_{d,\mathcal{S}} S_{d,\mathcal{S}}^\top + \frac{C}{N^2} \cdot \eta \cdot |\mathcal{S}| \cdot \mathrm{I} \right)^\dagger Y_\mathcal{S} \\
&= A_{d,(\mathcal{T},\mathcal{S})} \left( A_{d,(\mathcal{S},\mathcal{S})} + \frac{C}{N^2} \cdot \eta \cdot |\mathcal{S}| \cdot \mathrm{I} \right)^\dagger Y_\mathcal{S}.
\end{aligned}
\tag{C.36}
$$

This finishes the proof. □

**Lemma C.4.** *Let $\tilde{A} = \mathbb{E}[A] - \tau 11^\top$, where the adjacency matrix $A$ is drawn from the stochastic block model in Section 4.1. Then, for any $\xi > 0$, if $\rho > \max\{\alpha, \beta\}$, $\min\{\alpha, \beta\} > \gamma$ and $\tau < \frac{p\alpha + r\beta + (pr - p - r)\gamma}{pr}$, we have*

$$
\tilde{A}_{d,(\mathcal{T},\mathcal{S})} \left( \tilde{A}_{d,(\mathcal{S},\mathcal{S})} + \xi \mathrm{I} \right)^\dagger Y_\mathcal{S} = \frac{\lambda_\dagger}{\lambda_\dagger + p\xi} \cdot Y_\mathcal{T},
\tag{C.37}
$$

*where $d = r + p - 1$ and $\lambda_\dagger \triangleq n\rho - n\beta + n(p-1)\alpha - n(p-1)\gamma$. Furthermore, if we use $\tilde{\lambda}_i$ to denote the $i$-th largest eigenvalue of $\tilde{A}$, we have $\tilde{\lambda}_d - \tilde{\lambda}_{d+1} = n \cdot \min\{r(\beta - \gamma), p(\alpha - \gamma)\}$ and $\tilde{\lambda}_1 \leq np \cdot r\rho$.*

*Proof.* By the definition of the stochastic block model, every entry $\tilde{A}_{xx'}$ is in the set of $\{\rho - \tau, \alpha - \tau, \beta - \tau, \gamma - \tau\}$, depending on whether $x$ and $x'$ belong to the same domain/class. We can index every node $x$ as $(d_x, y_x, \mathrm{id}_x)$, where $\mathrm{id}_x \in [n]$ is the index of $x$ within domain $d_x$ and class $y_x$. For any integer $i \geq 1$, we use $1_i$ to denote the $i$-dimensional all-one vectors, and $\bar{1}_i = 1_i / \|1_i\|$ be its normalized unit vector. We use $\mathbb{S}^{i-1}$ to denote the $i$-dimensional unit-norm sphere.

It can be verified that $\tilde{A}$ can be decomposed into the sum of several matrix Kronecker products:

$$
\begin{aligned}
\tilde{A} =\; &(\beta - \gamma) \cdot \mathrm{I}_p \otimes (1_r 1_r^\top) \otimes (1_n 1_n^\top) \\
&+ (\alpha - \gamma) \cdot (1_p 1_p^\top) \otimes \mathrm{I}_r \otimes (1_n 1_n^\top) \\
&+ (\rho - \beta - \alpha + \gamma) \cdot \mathrm{I}_p \otimes \mathrm{I}_r \otimes (1_n 1_n^\top) \\
&+ \gamma \cdot (1_p 1_p^\top) \otimes (1_r 1_r^\top) \otimes (1_n 1_n^\top) \\
&- \tau \cdot (1_p 1_p^\top) \otimes (1_r 1_r^\top) \otimes (1_n 1_n^\top).
\end{aligned}
\tag{C.38}
$$

As a result, $\tilde{A}$ has the following four sets of eigenvectors with non-zero eigenvalues:

- $\bar{1}_p \otimes \bar{1}_r \otimes \bar{1}_n$. The corresponding eigenvalue is $\lambda_\circ \triangleq n\rho + n(r-1)\beta + n(p-1)\alpha + n(p-1)(r-1)\gamma - npr\tau$.

- $u \otimes \bar{1}_r \otimes \bar{1}_n$, where $u \in \mathbb{S}^{p-1}$ and $u^\top 1_p = 0$. The corresponding eigenvalue is $\lambda_\diamond \triangleq n\rho - n\alpha + n(r-1)\beta - n(r-1)\gamma$.

- $\bar{1}_p \otimes v \otimes 1_n$, where $v \in \mathbb{S}^{r-1}$ and $v^\top 1_r = 0$. The corresponding eigenvalue is $\lambda_\dagger \triangleq n\rho - n\beta + n(p-1)\alpha - n(p-1)\gamma$.

- $u \otimes v \otimes 1_n$, where $u \in \mathbb{S}^{p-1}$, $v \in \mathbb{S}^{r-1}$, $u^\top 1_p = 0$, and $v^\top 1_r = 0$. The corresponding eigenvalue is $\lambda_\ddagger \triangleq n\rho - n\beta - n\alpha + n\gamma$.

Since $\rho > \max\{\beta, \alpha\}$, $\min\{\alpha, \beta\} > \gamma$ and $\tau < \frac{p\alpha + r\beta + (pr - p - r)\gamma}{pr}$, we know that

$$\min\{\lambda_\circ, \lambda_\diamond, \lambda_\dagger, \} > \lambda_\ddagger, \tag{C.39}$$

and all these eigenvalues are positive. When $d = r + p - 1$, $\tilde{A}_d$ will contain exactly the first three sets of eigenvectors, since they correspond to the top-$d$ eigenvalues. This suggests that we can write $\tilde{A}_d$ as follows

$$\tilde{A}_d = \lambda_\circ \cdot \bar{1}_p \bar{1}_p^\top \otimes \bar{1}_r \bar{1}_r^\top \otimes \bar{1}_n \bar{1}_n^\top + \lambda_\diamond \cdot (I_p - \bar{1}_p \bar{1}_p^\top) \otimes \bar{1}_r \bar{1}_r^\top \otimes \bar{1}_n \bar{1}_n^\top + \lambda_\dagger \bar{1}_p \bar{1}_p^\top \otimes (I_r - \bar{1}_r \bar{1}_r^\top) \otimes \bar{1}_n \bar{1}_n^\top.$$

Restricting to the source domain, we have

$$\tilde{A}_{d,(\mathcal{S},\mathcal{S})} = \frac{\lambda_\circ + (p-1)\lambda_\diamond}{p} \cdot \bar{1}_r \bar{1}_r^\top \otimes \bar{1}_n \bar{1}_n^\top + \frac{\lambda_\dagger}{p} \cdot (I_r - \bar{1}_r \bar{1}_r^\top) \otimes \bar{1}_n \bar{1}_n^\top. \tag{C.40}$$

By the definition of pseudoinverse, we have

$$\left(\tilde{A}_{d,(\mathcal{S},\mathcal{S})} + \xi I\right)^\dagger = \left(\frac{\lambda_\circ + (p-1)\lambda_\diamond}{p} + \xi\right)^{-1} \cdot \bar{1}_r \bar{1}_r^\top \otimes \bar{1}_n \bar{1}_n^\top + \left(\frac{\lambda_\dagger}{p} + \xi\right)^{-1} \cdot (I_r - \bar{1}_r \bar{1}_r^\top) \otimes \bar{1}_n \bar{1}_n^\top.$$

Notice that $Y_\mathcal{S}$ satisfies $(\bar{1}_r \bar{1}_r^\top \otimes \bar{1}_n \bar{1}_n^\top) Y_\mathcal{S} = 0$ and $((I_r - \bar{1}_r \bar{1}_r^\top) \otimes \bar{1}_n \bar{1}_n^\top) Y_\mathcal{S} = Y_\mathcal{S}$, we have

$$(\tilde{A}_{d,(\mathcal{S},\mathcal{S})} + \xi I)^\dagger Y_\mathcal{S} = \left(\frac{\lambda_\dagger}{p} + \xi\right)^{-1} Y_\mathcal{S}. \tag{C.41}$$

We can also write $\tilde{A}_{d,(\mathcal{X},\mathcal{S})}$ in the form of Kronecker products as follows:

$$\tilde{A}_{d,(\mathcal{X},\mathcal{S})} = \frac{\lambda_\circ}{p} \cdot 1_p \otimes \bar{1}_r \bar{1}_r^\top \otimes \bar{1}_n \bar{1}_n^\top + \lambda_\diamond (e_1 - \frac{1}{p} 1_p) \otimes \bar{1}_r \bar{1}_r^\top \otimes \bar{1}_n \bar{1}_n^\top + \frac{\lambda_\dagger}{p} 1_p \otimes (I_r - \bar{1}_r \bar{1}_r^\top) \otimes \bar{1}_n \bar{1}_n^\top.$$

Again, using the fact that $(\bar{1}_r \bar{1}_r^\top \otimes \bar{1}_n \bar{1}_n^\top) Y_\mathcal{S} = 0$ and $((I_r - \bar{1}_r \bar{1}_r^\top) \otimes \bar{1}_n \bar{1}_n^\top) Y_\mathcal{S} = Y_\mathcal{S}$, we have

$$\tilde{A}_{d,(\mathcal{X},\mathcal{S})} \left(\tilde{A}_{d,(\mathcal{S},\mathcal{S})} + \xi I\right)^\dagger Y_\mathcal{S} = \frac{\lambda_\dagger}{\lambda_\dagger + p\xi} 1_p \otimes Y_\mathcal{S}. \tag{C.42}$$

Finally, noticing that $1_p \otimes Y_\mathcal{S} = Y$ finishes the proof. $\qquad \square$

**Lemma C.5.** *Let $A_d$ and $\tilde{A}_d$ be the rank-d approximations of $A$ and $\tilde{A}$, respectively. Let $\tilde{\lambda}_i$ be the i-th largest eigenvalue of $\tilde{A}$, $\|\cdot\|$ be the operator norm of a matrix or $\ell_2$-norm of a vector. Then when $\|A - \tilde{A}\| < \tilde{\lambda}_d - \tilde{\lambda}_{d+1}$, we have*

$$\|A_d - \tilde{A}_d\| \le \left(1 + \frac{2\|A - \tilde{A}\| + 2\|\tilde{A}\|}{(\tilde{\lambda}_d - \tilde{\lambda}_{d+1}) - \|A - \tilde{A}\|}\right) \cdot \|A - \tilde{A}\|. \tag{C.43}$$

*Proof.* Please see the proof in Shen et al. [44, Lemma 3]. $\qquad \square$

**Lemma C.6** (Theorem 5.2 in [31]). *Let $A$ be the adjacency matrix of a random graph on $N$ nodes in which edges occur independently. Let $\mathbb{E}[A] = \tilde{A}$ be the expectation adjacency matrix and assume that $N \max_{i,j} \tilde{A}_{ij} \ge \log N$. Then, for any $\xi > 0$, there exists a constant $C = C(\xi)$ such that*

$$\|A - P\| \le C\sqrt{N} \tag{C.44}$$

*with probability at least $1 - N^{-2\xi}$.*

### C.3 PROOF OF THEOREM 3.3

**Theorem 3.3** *The global minimum of the supervised spectral contrastive loss $\mathcal{L}^{sup}(f;\tau)$ in Equation 3.8 is uniquely obtained at $f(x) = \hat{h}_c$, for any $i, c \in [r]$ and $x \sim \mathcal{A}(x_{i,c})$, where $\hat{H} = [\hat{h}_1, ..., \hat{h}_r]$ is the minimizer of $\left\| (r\mathrm{I} - \tau 11^\top) - H^\top H \right\|_F^2$. More specifically, if $r \leq d+1$, $\hat{H}$ satisfies $\hat{H}^\top \hat{H} = r\mathrm{I} - \tau 11^\top$; if $r > d+1$, we have $\hat{H}^\top \hat{H} = \mathcal{P}_d(r\mathrm{I} - \tau 11^\top)$, where $\mathcal{P}_d(X)$ denotes the best $d$-rank approximation of $X$.*

*Proof.* Let $h_{i,c} = \mathbb{E}_{x \sim \mathcal{A}(x_{i,c})}[f(x)]$, $h_c = \frac{1}{n}\sum_{i=1}^n h_{i,c}$, and $H = [h_1, h_2, ..., h_r]$.

For the positive part $\mathcal{L}_{pos}^{sup}$, we have

$$
\begin{aligned}
\mathcal{L}_{pos}^{sup} &= -\frac{2}{rn^2} \sum_{i=1}^n \sum_{j=1}^n \sum_{c=1}^r \left( \mathbb{E}_{x \sim \mathcal{A}(x_{i,c})}[f(x)] \right)^\top \left( \mathbb{E}_{x^+ \sim \mathcal{A}(x_{j,c})}[f(x^+)] \right) \\
&= -\frac{2}{rn^2} \sum_{c=1}^r \left( \sum_{i=1}^n \mathbb{E}_{x \sim \mathcal{A}(x_{i,c})}[f(x)] \right)^\top \left( \sum_{j=1}^n \mathbb{E}_{x^+ \sim \mathcal{A}(x_{j,c})}[f(x^+)] \right) \\
&= -\frac{2}{rn^2} \sum_{c=1}^r \left( \sum_{i=1}^n h_{i,c} \right)^\top \left( \sum_{j=1}^n h_{j,c} \right) \\
&= -\frac{2}{r} \sum_{c=1}^r \| h_c \|_2^2
\end{aligned}
\tag{C.45}
$$

For the negative part $\mathcal{L}_{neg}$, we have

$$
\begin{aligned}
\mathcal{L}_{neg} &= \frac{1}{r^2 n^2} \sum_{i=1}^n \sum_{j=1}^n \sum_{c=1}^r \sum_{k=1}^r \mathbb{E}_{\substack{x \sim \mathcal{A}(x_{i,c}) \\ x^- \sim \mathcal{A}(x_{j,k})}} \left[ \left( f(x)^\top f(x^-) + \tau \right)^2 \right] \\
&\geq \frac{1}{r^2} \sum_{c=1}^r \sum_{k=1}^r \left[ \frac{1}{n^2} \sum_{i=1}^n \sum_{j=1}^n \mathbb{E}_{\substack{x \sim \mathcal{A}(x_{i,c}) \\ x^- \sim \mathcal{A}(x_{j,k})}} \left[ f(x)^\top f(x^-) + \tau \right] \right]^2 \\
&= \frac{1}{r^2} \sum_{c=1}^r \sum_{k=1}^r \left( h_c^\top h_k + \tau \right)^2,
\end{aligned}
\tag{C.46}
$$

where the equality holds if and only if $f(x)^\top f(x^-)$ is a constant for any $i, j \in [r]$, $x \sim \mathcal{A}(x_{i,c})$, and $x^- \sim \mathcal{A}(x_{j,k})$.

Combining Equation C.45 and Equation C.46, we have

$$
\begin{aligned}
\mathcal{L}^{sup}(f;\tau) &\geq -\frac{2}{r} \sum_{c=1}^r \| h_c \|_2^2 + \frac{1}{r^2} \sum_{c=1}^r \sum_{k=1}^r (h_c^\top h_k + \tau)^2 \\
&= \frac{1}{r^2} \left\| (r\mathrm{I} - \tau 11^\top) - H^\top H \right\|_F^2 + const
\end{aligned}
\tag{C.47}
$$

According to Eckart-Young-Mirsky theorem [13], the global minimizer $\hat{H}$ of $\left\| (r\mathrm{I} - \tau 11^\top) - H^\top H \right\|_F^2$ is the best $d$-rank approximation of $r\mathrm{I} - \tau 11^\top$.

More specifically, if $r \leq d+1$, $\hat{H}$ satisfies $\hat{H}^\top \hat{H} = r\mathrm{I} - \tau 11^\top$; if $r > d+1$, we have $\hat{H}^\top \hat{H} = \mathcal{P}_d(r\mathrm{I} - \tau 11^\top)$, where $\mathcal{P}_d(X)$ denotes the best $d$-rank approximation of $X$. □

### C.4 PROOF OF THEOREM 3.4

**Theorem 3.4** *For the noise transition matrix $W$ defined above, the global minimum of $\mathcal{L}_{noise}^{sup}(f;\tau)$ is uniquely obtained at $\forall i \in [n], \forall c \in [r], \forall x \sim \mathcal{A}(x_{i,c})$, $f(x) = \hat{h}_c$, where $\hat{H} = [\hat{h}_1, ..., \hat{h}_r]^\top$ is the minimizer of $\left\| (rW - \tau 11^\top) - H^\top H \right\|_F^2$.*

*Proof.* According to the definition of the label noise model, we know that

$$\mathcal{L}_{noise}^{sup}(f;\tau) = \mathcal{L}_{noise,pos}^{sup}(f;\tau) + \mathcal{L}_{noise,neg}^{sup}(f;\tau) \tag{C.48}$$

where

$$
\begin{aligned}
\mathcal{L}_{noise,pos}^{sup}(f;\tau) &= -\frac{2}{cn^2}\sum_{i=1}^{n}\sum_{j=1}^{n}\sum_{c=1}^{r}\left(\mathbb{E}_{x\sim\mathcal{A}(x_{i,c})}[f(x)]\right)^{\top}\left(\sum_{c'=1}^{r}w_{cc'}\mathbb{E}_{x^+\sim\mathcal{A}(x_{j,c'})}f(x^+)\right)\\
&= -\frac{2}{r}\sum_{c=1}^{r}h_c^{\top}\left(\sum_{c'=1}^{r}w_{cc'}h_c'\right) = -\frac{2}{r}\sum_{c=1}^{r}\sum_{c'=1}^{r}w_{cc'}h_c^{\top}h_{c'}
\end{aligned}
\tag{C.49}
$$

and

$$
\begin{aligned}
\mathcal{L}_{noise,neg}^{sup}(f;\tau) &= \frac{1}{r^2n^2}\sum_{i=1}^{n}\sum_{j=1}^{n}\sum_{c=1}^{r}\sum_{k=1}^{r}\sum_{c'=1}^{r}w_{kc'}\mathbb{E}_{\substack{x\sim\mathcal{A}(x_{i,c})\\x^-\sim\mathcal{A}(x_{j,c'})}}\left[\left(f(x)^{\top}f(x^-)+\tau\right)^2\right]\\
&= \frac{1}{rn^2}\sum_{i=1}^{n}\sum_{j=1}^{n}\sum_{c=1}^{r}\sum_{c'=1}^{r}\sum_{k=1}^{r}w_{kc'}\mathbb{E}_{\substack{x\sim\mathcal{A}(x_{i,c})\\x^-\sim\mathcal{A}(x_{j,c'})}}\left[\left(f(x)^{\top}f(x^-)+\tau\right)^2\right]\\
&= \frac{1}{rn^2}\sum_{i=1}^{n}\sum_{j=1}^{n}\sum_{c=1}^{r}\sum_{c'=1}^{r}\mathbb{E}_{\substack{x\sim\mathcal{A}(x_{i,c})\\x^-\sim\mathcal{A}(x_{j,c'})}}\left[\left(f(x)^{\top}f(x^-)+\tau\right)^2\right]\\
&\geq \frac{1}{r^2}\sum_{c=1}^{r}\sum_{c'=1}^{r}(h_c^{\top}h_{c'}+\tau)^2,
\end{aligned}
\tag{C.50}
$$

where the second equality is based on the fact that $\sum_{k=1}^{r}w_{kc'}=1$. We further have

$$\mathcal{L}_{noise}^{sup}(f;\tau) \geq \frac{1}{r^2}\left\|(rW-\tau11^{\top})-H^{\top}H\right\|_F^2 + const. \tag{C.51}$$

This finishes the proof. $\qquad\square$

## C.5 PROOF OF PROPOSITION 3.5

**Proposition 3.5** *Considering the symmetric label noise [15] in which $w_{cc'}=1-(r-1)\omega$ for $c=c'$, $w_{cc'}=\omega$ for $c\neq c'$, and $\omega < \frac{1}{r}$. If $\tau \geq r\omega$, let $\hat{f}=\arg\min_f \mathcal{L}_{noise}^{sup}(f;\tau)$, then $\frac{1}{\sqrt{1-r\omega}}\cdot\hat{f}$ is also the global minimizer of $\mathcal{L}^{sup}\left(f;\frac{\tau-r\omega}{1-r\omega}\right)$.*

*Proof.* The symmetric label noise model means that $W=(1-r\omega)I+\omega11^{\top}$, according to Theorem 3.4, we know that $\forall i\in[n], \forall c\in[r], \forall x\sim\mathcal{A}(x_{i,c}), \hat{f}(x)=\hat{h}_c$, where $\hat{H}=[\hat{h}_1,...,\hat{h}_r]$ is the minimizer of

$$
\begin{aligned}
&\left\|(rW-\tau11^{\top})-H^{\top}H\right\|_F^2\\
&= \left\|(r(1-r\omega)I-(\tau-r\omega)11^{\top})-H^{\top}H\right\|_F^2\\
&= (1-r\omega)^2\left\|(rI-\tfrac{\tau-r\omega}{1-r\omega}11^{\top})-\left(\tfrac{1}{\sqrt{1-r\omega}}\cdot H\right)^{\top}\left(\tfrac{1}{\sqrt{1-r\omega}}\cdot H\right)\right\|_F^2,
\end{aligned}
\tag{C.52}
$$

thus, $\frac{1}{\sqrt{1-r\omega}}\cdot\hat{f}(x)$ is also the minimizer of $\mathcal{L}^{sup}(f;\frac{\tau-r\omega}{1-r\omega})$ in accordance with Theorem 3.3. $\qquad\square$

