# OpenReview forum: "Zero-Mean Regularized Spectral Contrastive Learning: Implicitly Mitigating Wrong Connections in Positive-Pair Graphs"
_ICLR.cc/2024/Conference — ICLR 2024 poster_

### Official Review · Reviewer_41KY · 2023-10-28

**Soundness:** 3 good
**Presentation:** 3 good
**Contribution:** 2 fair
**Rating:** 6
**Confidence:** 3

**Summary:**

By incorporating an additive factor into the SpeCL term that involves negative pairs, this paper enforces the mean of representations to be zero. The experimental results and related theoretical analysis suggest that introducing $\tau$ can improve performance.

**Strengths:**

1.This paper addresses the issue of negative pairs in contrastive learning, which is a hot topic in the research community. The authors aim to provide a solution to this problem that arises in related CL works.

2.The authors of this paper present a theoretical foundation to support the proposed zero-mean regularization in both the unsupervised domain adaptation (UDA) task and supervised classification with noisy labels.

3.The proposed method's robustness is improved by the impressive results achieved on the supervised classification task, particularly when dealing with symmetric label noise.

**Weaknesses:**

1. This work appears to be relatively incremental based on Ref. [18] and [19]. Basically, what caught my interest is the representation of spectral embeddings using affinities and features; however, it was previously proposed in [18].

2. Although the authors describe the proposed additive factor as simple yet effective, this work can still be considered progressive research. However, when compared to [18] and [19], it is unclear whether significant progress has been made for this conference.

3. The effectiveness of the proposed method is demonstrated through experiments that involve fewer state-of-the-art methods.

**Questions:**

1. On page 4, there is a missing space in "thatSpeCL".

2. It is suggested to include more benchmark methods in the experiments to avoid categorizing it as an enhanced/improved version of SpeCL method, and instead position it as a robust SSL method.

3. The factor $τ$ effectively relaxes the orthogonal constraint on negative pairs. It would be interesting to explore the integration of other methods, such as Barlow Twin, in different ways.

4. In the experiments, it is recommended to analyze the results achieved with different values of $\tau$ due to its previous detailed analysis. This analysis can provide insights into what can be inferred from the results obtained with different $\tau$ values.

---

> ### Author Response · Authors · 2023-11-19
> **Response to Reviewer Reviewer 41KY (1/2)**
>
> We thank the reviewer for the constructive comments and insightful suggestions.
>
> **Comment 1:** *This work appears to be relatively incremental based on Ref. [18] and [19]. Basically, what caught my interest is the representation of spectral embeddings using affinities and features; however, it was previously proposed in [18].* and *Although the authors describe the proposed additive factor as simple yet effective, this work can still be considered progressive research. However, when compared to [18] and [19], it is unclear whether significant progress has been made for this conference.*
>
> **Response:** Thank you for your kind comment.
> At first glance, the formulation of the proposed zero-mean regularized spectral contrastive loss (ZM-SpeCL) may appear similar to the vanilla SpeCL [18] and [19], with the only difference being the introduction of $\tau$ in the term involving negative pair. However, it cannot be simply justified that our work is an incremental improvement over SpeCL. Otherwise, following the same logic, some recent well-known works in self-supervised learning, such as BYOL and SimSiam, could also be considered incremental compared to SimCLR.
>
> More importantly, ZM-SpeCL possesses a clear theoretical motivation and support. It is specifically designed to modify pairwise similarities between positive pairs, which intuitively relaxes the
> orthogonality of representations between negative pairs and implicitly alleviates
> the adverse effect of wrong connections in the positive-pair graph, leading to better
> performance and robustness. Our work is not only simple yet effective, but also theoretically sound.
>
> In Section 3.2, we delve into the role of zero-mean regularization on spectral contrastive pretraining-based unsupervised domain adaptation and provide theoretical proof that zero-mean regularization can tighten the error bound by multiplying a factor less than one. In Section 3.3, we establish the supervised version of spectral contrastive loss and derive the closed-form optimal representations, which resembles the Neural Collapse phenomenon and suggests using class-mean features as a classifier. We further prove that zero-mean regularization can mitigate label noise by implicitly reducing mislabeled weights in the noise transition matrix.
>
> ------
>
> **Comment 2:** *The effectiveness of the proposed method is demonstrated through experiments that involve fewer state-of-the-art methods.* and *It is suggested to include more benchmark methods in the experiments to avoid categorizing it as an enhanced/improved version of SpeCL method, and instead position it as a robust SSL method.*
>
> **Response:** Thank you for your constructive comment. In response to your suggestion, we have expanded our experimental evaluation to include comparisons with additional state-of-the-art methods, We have incorporated self-supervised learning methods such as MoCo, SimCLR, and BYOL into our comparative analysis. For these experiments, we employed a PreAct-ResNet18 architecture and conducted training over a total of 200 epochs. The training process utilized the SGD optimizer with a momentum of 0.9, a learning rate of 0.1, and a weight decay of 5e-4. To dynamically adjust the learning rate over the course of the 200 epochs, we implemented the cosine decay learning rate schedule.
>
> The comprehensive experimental results, including comparisons with the aforementioned state-of-the-art methods, are presented in the table below. We believe that this expanded evaluation provides a more thorough understanding of the effectiveness of our proposed method in the context of self-supervised learning.
>
> | Method             | CIFAR-10  | CIFAR-100 | SVHN      |
> | :----------------- | :-------- | :-------- | :-------- |
> | SimCLR             | 83.84     | 55.04     | 88.07     |
> | MoCo               | 73.63     | 51.88     | 82.25     |
> | BYOL               | 86.32     | 54.73     | 90.24     |
> | SpeCL              | 86.25     | 52.44     | 89.58     |
> | SpeCL ($\tau$=0.1) | 86.77     | 53.98     | 90.36     |
> | SpeCL ($\tau$=0.2) | 86.93     | 55.25     | 90.80     |
> | SpeCL ($\tau$=0.5) | 87.05     | 56.61     | 91.10     |
> | SpeCL ($\tau$=1.0) | **87.24** | **56.77** | **91.12** |
>
> ------
>
> **Comment 3:** On page 4, there is a missing space in "thatSpeCL".
>
> **Response:** Thank you for your careful review. We have rectified the mentioned typos and addressed some other potential errors.

---

> ### Author Response · Authors · 2023-11-19
> **Response to Reviewer Reviewer 41KY (2/2)**
>
> **Comment 4:** The factor $\tau$ effectively relaxes the orthogonal constraint on negative pairs. It would be interesting to explore the integration of other methods, such as Barlow Twin, in different ways.
>
> **Response:** Thank you for your insightful comment and suggestion. We have explored the integration of other methods, specifically Barlow Twins, by incorporating the relaxation factor $\tau$ to effectively relax the orthogonal constraint on negative pairs. The modified Barlow Twins loss function, denoted as $L_{BT}$, is expressed as follows:
> \begin{equation}
>     L_{BT}=\sum_i (1-C_ii)^2 + \lambda \sum_{i\neq j} (C_{ij}+\tau)^2,
> \end{equation}
> where $C$ is the cross-correlation matrix computed between the outputs of two batches of samples. The hyperparameter $\tau$ is introduced to control the relaxation of the orthogonality constraint.
>
> To provide empirical evidence of the impact of incorporating $\tau$ into Barlow Twins, we trained the models for 200 epochs using the open-source code available at https://github.com/IgorSusmelj/barlowtwins. The results, presented in the table below, demonstrate the KNN accuracy for different values of $\tau$:
>
> | $\tau$       | 0.0   | 0.1   | 0.2   | 0.5   | 1.0   |
> | ------------ | ----- | ----- | ----- | ----- | ----- |
> | KNN Accuracy | 74.63 | 74.99 | 74.89 | 74.87 | 74.20 |
>
> The results indicate that the incorporation of $\tau$ in Barlow Twins improves downstream KNN classification accuracy. We appreciate your suggestion, and these findings suggest that the introduced relaxation factor can positively influence the performance of Barlow Twins in the context of our experiments.
>
>
> \hspace*{\fill}
>
> \noindent\textbf{Comment 5:} In the experiments, it is recommended to analyze the results achieved with different values of $\tau$ due to its previous detailed analysis. This analysis can provide insights into what can be inferred from the results obtained with different values.
>
> \noindent\textbf{Response:} We appreciate your valuable feedback and would like to provide additional insights into the role of the regularization parameter $\tau$ and the trade-offs associated with its selection. In this paper, we propose setting $\tau\in[0,1]$ to balance the regularization strength of zero-mean regularization. Although in Section 3, we do not explicitly constrain $\tau\le 1$, theoretical and empirical evidence supports the necessity of such a constraint:
>
>
> - In Theorem 3.1, we assume that $\tau<\frac{\tilde{\lambda}_1(0)-\tilde{\lambda}_d(0)}{\tilde{\lambda}_1(0)}<1$ to ensure that the introduction of $\tau$ will not alter the value of the $d$-th eigenvalue in Eq. (C.23). In Proposition 3.2, we assume that $\tau\le \frac{\tilde{\lambda}_1(0)-\tilde{\lambda}_2(0)}{\tilde{\lambda}_1(0)}< \frac{\tilde{\lambda}_1(0)-\tilde{\lambda}_d(0)}{\tilde{\lambda}_1(0)}<1$ to guarantee that the first eigenvalue of $\tilde{A}'$ is $\tilde{\lambda}_1(\tau)=(1-\tau)\tilde{\lambda}_1(0)$ (as can be seen, $\tau\le 1$ is essential to avoid the presence of negative eigenvalues of $\tilde{A}'$) in Eq. (C.32), thus facilitating a more intuitive comparison with the bound at $\tau=0$. While it is possible to draw conclusions within the range of $[\frac{\tilde{\lambda}_1(0)-\tilde{\lambda}_d(0)}{\tilde{\lambda}_1(0)}, 1]$, it requires a more nuanced comparison of the order of these eigenvalues, predictably resulting in an obviously different bound than at $\tau=0$.
>
> - In Theorem 3.3, we prove that the global minimum of the supervised spectral contrastive loss is achieved at $\hat{H}=\arg\min_{H}\|(rI-\tau11^\top)-H^\top H\|_F^2$. When $r\le d+1$, $\|(rI-\tau11^\top)-H^\top H\|_F^2$ has a minimum of zero only for $\tau\in [0,1]$, corresponding to $\hat{H}^\top \hat{H}=rI-\tau11^\top$, where the angle between representations of different classes is $\arccos\frac{-\tau}{r-\tau}>90^\circ$. However, for $\tau > 1$, there exists no $\hat{H}$ that satisfies $\hat{H}^\top \hat{H}=rI-\tau11^\top$, as it would lead to a paradoxical situation where $0\le 1^\top\hat{H}^\top \hat{H} 1= 1^\top(rI-\tau 11^\top)1=r^2(1-\tau)<0$.
>
> - We conducted experiments for $\tau>1$ as presented in Tables 9 and 11 of the Appendix. As observed, consistently better results are obtained with $\tau\in[0,1]$, while excessively large $\tau$ values hinder overall performance. Thus, we suggest limiting $\tau$ to the range of $[0,1]$ to balance the regularization and discriminative capabilities.
>
> Many thanks to the reviewer for the valuable suggestion, we have added this part to Section A.4 in the revised version.

---

> ### Author Response · Authors · 2023-11-23
> **Looking forward to your feedback**
>
> Dear reviewer 41KY,
>
> Thanks again for your valuable time and insightful comments. As the deadline for the Author/Reviewer discussion is approaching, it would be nice of you to let us know whether our answers have solved your concerns so that we can better improve our work. We are looking forward to your feedback.
>
> Best regards,
>
> Authors of Paper 224

---

### Official Review · Reviewer_9yLv · 2023-10-31

**Soundness:** 3 good
**Presentation:** 3 good
**Contribution:** 3 good
**Rating:** 6
**Confidence:** 3

**Summary:**

The authors provide a new regularizer for (spectral) contrastive
learning which supposedly improves the quality of the representation.
The authors provide experiments to show the quantitative improvements
due to their method.

**Strengths:**

The regularization is well-motivated and intuitively explained.  The
experiments seem convincing and suggest that the improvement is
consisent across datasets.

The appendix is extensive.

**Weaknesses:**

The results on the CIFAR datasets are not state-of-the-art for
contrastive learning.  This begs the question whether the regularizer
could also be applied to other CL techniques such as SimCLR or others.
Hopefully the regularization would also help in that case.

Table 1 & 2 only report single numbers.  It would be more convincing
if there was a mean +- std reported or some other statisic computed
over multiple runs.

**Questions:**

Could the approach be extended to other CL techniques?  Do you expect
it to improve the results similarly?

---

> ### Author Response · Authors · 2023-11-19
> **Response to Reviewer 9yLv (1/2)**
>
> We thank the reviewer for the constructive comments and insightful suggestions.
>
> **Comment 1**: The results on the CIFAR datasets are not state-of-the-art for contrastive learning. This begs the question whether the regularizer could also be applied to other CL techniques such as SimCLR or others. Hopefully the regularization would also help in that case.
>
> **Response**: Thank you for your insightful comment.  We would like to provide further clarification on the applicability of the zero-mean regularization to other contrastive learning (CL) techniques.
>
> The primary concept behind zero-mean regularization is to implicitly reduce the weights of incorrect connections in the positive-pair graph, as emphasized throughout the work. This idea is fully depicted in the unsupervised domain adaptation (see Eq. 3.3) and learning with noisy labels (see Theorem 3.4) in Section 3.3.2. Our focus in this work has been on the spectral contrastive loss due to its theoretical soundness and ease of analysis. The manifestation is succinctly expressed as the loss-specific "zero-mean regularization".
>
> To extend the core idea of ``weight reduction in positive pairs" to other popular CL techniques, such as SimCLR, MoCo, and CPC, we propose a modification to the InfoNCE loss by uniformly reducing the importance of positive pair as follows:
> \begin{equation}
> -\log \frac{(1-\tau)e^{sim(x,x^+)}}{(1-\tau)e^{sim(x,x^+)}+\sum_{i=1}^ke^{sim(x,x_i^-)}}
> =-\log \frac{e^{sim(x,x^+)-\tau'}}{e^{sim(x,x^+)-\tau'}+\sum_{i=1}^ke^{sim(x,x_i^-)}}
> =-\log \frac{e^{sim(x,x^+)}}{e^{sim(x,x^+)}+\sum_{i=1}^ke^{sim(x,x_i^-)+\tau'}},
> \end{equation}
> where $sim(x_1,x_2)$ denotes the similarity between $x_1$ and $x_2$, $\tau'=\log\frac{1}{1-\tau}>0$ and $\tau\in[0, 1)$. As can be seen, the derived form that adds a margin term is similar to margin-based losses, particularly the negative-margin softmax loss [1]. From this point of view, weight reduction in positive pairs coincides with the motivation of margin-based losses  that enlarges the discriminativeness with intuitive decision boundaries.
>
> For empirical validation of the efficacy of the modified InfoNCE loss, we conducted experiments on self-supervised and supervised learning using CIFAR-10/-100 and SVHN datasets. These additional experiments aim to provide further evidence of the versatility inherent in the concept of "weight reduction in wrong positive pairs" and demonstrate its potential to enhance the performance of various contrastive learning algorithms on real-world datasets.
> **Self-Supervised Learning:**
>
> |                          | CIFAR-10  | CIFAR-100 | SVHN       |
> | :----------------------- | :-------- | :-------- | :--------- |
> | infoNCE                  | 86.56     | 55.51     | 90.769     |
> | infoNCE with $\tau=0.05$ | **86.64** | **55.68** | **91.049** |
> | infoNCE with $\tau=0.1$  | **86.97** | **56.07** | **90.934** |
> | infoNCE with $\tau=0.2$  | **86.65** | **55.55** | 90.769     |
> | infoNCE with $\tau=0.5$  | **87.32** | **56.01** | **90.953** |
> | infoNCE with $\tau=0.7$  | 86.41     | **55.98** | **90.896** |
> | infoNCE with $\tau=0.9 $ | 86.48     | **55.88** | **90.953** |
> | infoNCE with $\tau=0.95$ | **86.80** | **55.56** | **91.288** |
>
> **Supervised Contrastive Learning**:
>
> |                          | CIFAR-10  | CIFAR-100 | SVHN       |
> | :----------------------- | :-------- | :-------- | :--------- |
> | infoNCE                  | 94.18     | 71.43     | 96.105     |
> | infoNCE with $\tau=0.05$ | 94.04     | **71.80** | **96.235** |
> | infoNCE with $\tau=0.1$  | **94.27** | **71.56** | **96.166** |
> | infoNCE with $\tau=0.2$  | **94.19** | **72.07** | **96.201** |
> | infoNCE with $\tau=0.5$  | 94.17     | **72.44** | **96.182** |
> | infoNCE with $\tau=0.7$  | **94.54** | **72.52** | **96.136** |
> | infoNCE with $\tau=0.9 $ | **94.43** | **72.29** | **96.155** |
> | infoNCE with $\tau=0.95$ | **94.33** | **71.82** | **96.097** |
>
> > [1] Bin Liu, Yue Cao, Yutong Lin, Qi Li, Zheng Zhang, Mingsheng Long, and Han Hu. Negative margin matters: Understanding margin in few-shot classification. In Computer Vision–ECCV 2020: 16th European Conference, Glasgow, UK, August 23–28, 2020, Proceedings, Part IV 16, pages 438–455. Springer, 2020.
>
> -------

---

> ### Author Response · Authors · 2023-11-19
> **Response to Reviewer 9yLv (2/2)**
>
> **Comment 2:** Table 1 \& 2 only report single numbers. It would be more convincing if there was a mean +- std reported or some other statisic computed over multiple runs.
>
> **Response**: Thank you for your kind suggestion. In the revised manuscript, we have updated the comparison results by including mean values along with their corresponding standard deviations (mean $\pm$ std).
>
> **Self-Supervised Learning:**
>
> |                       | CIFAR-10             | CIFAR-100            | SVHN                 |
> | :-------------------- | :------------------- | :------------------- | :------------------- |
> | Vanilla SpeCL         | 86.22 $\pm$ 0.12     | 52.66 $\pm$ 0.16     | 89.77 $\pm$ 0.19     |
> | SpeCL with $\tau=0.1$ | **86.83 $\pm$ 0.12** | **53.97 $\pm$ 0.19** | **90.26 $\pm$ 0.07** |
> | SpeCL with $\tau=0.2$ | **86.84 $\pm$ 0.10** | **55.27 $\pm$ 0.02** | **90.78 $\pm$ 0.07** |
> | SpeCL with $\tau=0.5$ | **87.15 $\pm$ 0.12** | **56.37 $\pm$ 0.27** | **91.10 $\pm$ 0.13** |
> | SpeCL with $\tau=1.0$ | **87.09 $\pm$ 0.16** | **56.63 $\pm$ 0.10** | **91.24 $\pm$ 0.09** |
>
> **Supervised Contrastive Learning:**
>
> |                       | CIFAR-10             | CIFAR-100            | SVHN                 |
> | :-------------------- | :------------------- | :------------------- | :------------------- |
> | Vanilla SpeCL         | 93.47 $\pm$ 0.46     | 65.59 $\pm$ 0.75     | 96.05 $\pm$ 0.16     |
> | SpeCL with $\tau=0.1$ | **94.21 $\pm$ 0.13** | **68.15 $\pm$ 0.19** | **96.22 $\pm$ 0.05** |
> | SpeCL with $\tau=0.2$ | **94.52 $\pm$ 0.05** | **68.04 $\pm$ 0.18** | **96.28 $\pm$ 0.05** |
> | SpeCL with $\tau=0.5$ | **94.36 $\pm$ 0.12** | **69.06 $\pm$ 0.34** | **96.22 $\pm$ 0.09** |
> | SpeCL with $\tau=1.0$ | **94.38 $\pm$ 0.06** | **69.59 $\pm$ 0.11** | **96.25 $\pm$ 0.10** |

---

> ### Author Response · Authors · 2023-11-23
> **Looking forward to your feedback**
>
> Dear reviewer 9yLv,
>
> Thanks again for your valuable time and insightful comments. As the deadline for the Author/Reviewer discussion is approaching, it would be nice of you to let us know whether our answers have solved your concerns so that we can better improve our work. We are looking forward to your feedback.
>
> Best regards,
>
> Authors of Paper 224

---

> > ### Comment · Reviewer_9yLv · 2023-11-23
> >
> > Thank you for the response.  I appreciate the additional detail.  I will keep my score (as it already is positive).

---

### Official Review · Reviewer_Ef5A · 2023-10-31

**Soundness:** 4 excellent
**Presentation:** 4 excellent
**Contribution:** 2 fair
**Rating:** 6
**Confidence:** 5

**Summary:**

In this point, the authors try to extend spectral contrastive learning with negative pairs. By adding a zero-mean regularization, the loss function relaxes the orthogonality of representations between negative pairs and implicitly alleviates the adverse effect of wrong connections in the positive-pair graph, leading to better performance and robustness. Beyond that, this paper gives a solid theoretical analysis for the regularization.

**Strengths:**

1. This paper is very easy to follow. The motivation is very clear, and the methodology is elegant.

2. The perspective of UDA is novel. It provides a different way to observe contrastive learning.

3. The perspective of supervised classification with a noise label is also helpful for contrastive learning.

**Weaknesses:**

1. The novelty is an issue. According to the Equ 3.2, it is the same as contrastive laplacian eigenmaps (NeurIPS 2021). In contrastive laplacian eigenmaps, they have the same three terms. The main point is the fully connected adjacent matrix (the all-one matrix).

**Questions:**

I would like to hear from authors about the relationship between this one and contrastive laplacian eigenmaps. I think the authors give a totally different theoretical analysis based on a very similar thing.

---

> ### Author Response · Authors · 2023-11-19
> **Response to Reviewer Ef5A**
>
> We thank the reviewer for the constructive comments and insightful suggestions.
>
> **Comment:** *The novelty is an issue. According to the Equ 3.2, it is the same as contrastive laplacian eigenmaps (NeurIPS 2021). In contrastive laplacian eigenmaps, they have the same three terms. The main point is the fully connected adjacent matrix (the all-one matrix).* and *I would like to hear from authors about the relationship between this one and contrastive laplacian eigenmaps [1]. I think the authors give a totally different theoretical analysis based on a very similar thing.*
>
> **Response:** Thank you for your insightful comment. While there are certain similarities between Zero-mean regularized spectral loss (Zero-SpeCL) in Eq. 3.2 and contrastive Laplacian eigenmaps (COLES) [1], there are distinct differences between them:
>
> - `Different Negative Components.` The contrastive objective (to be maximized) can generally be formulated as $J(f)=E_{(x,x^+)\sim p_{pos}} s(f(x),f(x^+)) + \eta E_{(x,x^-)\sim p_{data}} \tilde{s}(f(x),f(x^-))$. While the positive components of Zero-SpeCL and COLES are the same, their negative components are different. The negative component in COLES is $\tilde{s}(f(x),f(x^-))=-f(x)^\top f(x^-)$, whereas in Zero-SpeCL, it is $\tilde{s}(f(x),f(x^-))=-(f(x)^\top f(x^-)+\tau)^2$.
> - `Different Reasons for Covariance Term.` Both Zero-SpeCL and COLES introduce the term $\Vert \mathbb{E}_x f(x) f(x)^\top-I\Vert_F^2$. The motivation of COLES is to softly satisfy the constraint $F^\top F=I$ that removes an arbitrary scaling factor in the embeddings [2]. Additionally, it helps avoid collapsed solutions since, without the constraint $F^\top F=I$, the derived graph Dirichlet energy $-Tr(F^\top\Delta W F)$ would be minimized when all representations collapse to a constant vector. In contrast, the covariance term $\mathcal{R}_1(f)=\Vert \mathbb{E}_x f(x) f(x)^\top-I\Vert_F^2$ in Zero-SpeCL appears as a part of an equivalent form in Eq. 3.2.
> - `Different Overall Objective.` COLES represents a constrained graph Dirichlet energy minimization problem (contrastive Laplacian eigenmaps), i.e.,  $\min_{F^\top F=I}-Tr(F^\top\Delta W F)$. On the other hand, Zero-SpeCL depicts the low-rank matrix approximation problem $\min_F|A-F^\top F|_F^2$ (as shown in Eq. 3.3).
> - `Different Focuses.` Though both works revolve around contrastive learning, our work pays more attention on discussing in theory how zero-mean regularization is helpful in downstream unsupervised domain adaptation (UDA) and supervised classification with noisy labels. We theoretically prove that zero-mean regularization can tighten the error bound by multiplying a factor less than one for UDA (Section 3.2), and prove that zero-mean regularization can mitigate label noise by implicitly reducing mislabeled weights in the noise transition matrix (Section 3.3).
>
> Thanks again to the reviewers for their comments. In order to better clarify our contribution, we have added the relevant comparison to contrastive laplacian eigenmaps in the revised version.
>
> >[1] Zhu H, Sun K, Koniusz P. Contrastive laplacian eigenmaps[J]. Advances in Neural Information Processing Systems, 2021, 34: 5682-5695.
> >
> > [2] Belkin M, Niyogi P. Laplacian eigenmaps and spectral techniques for embedding and clustering[J]. Advances in neural information processing systems, 2001, 14.

---

> ### Author Response · Authors · 2023-11-23
> **Looking forward to your feedback**
>
> Dear reviewer Ef5A,
>
> Thanks again for your valuable time and insightful comments. As the deadline for the Author/Reviewer discussion is approaching, it would be nice of you to let us know whether our answers have solved your concerns so that we can better improve our work. We are looking forward to your feedback.
>
> Best regards,
>
> Authors of Paper 224

---

### Author Response · Authors · 2023-11-23
**Appreciation for Your Review and Openness to Further Comments**

Dear Reviewer,

We sincerely appreciate your dedicated time and effort in reviewing our paper. Your insightful comments have been invaluable in enhancing the quality of our work. Should you have any additional comments or queries regarding our response, we would be delighted to address them. Your feedback is highly valuable to us, and we are committed to ensuring clarity and addressing any concerns you may have.

Thank you once again for your thorough review.

Best regards,

The Authors

---

### Meta-Review · Area_Chair_LctV · 2023-12-07

**Metareview:**

Thanks for your submission to ICLR.

This paper examines a technique for contrastive learning.  On the positive side, the reviewers noted that the paper is well-written, with lots of results (particularly in the appendix), and also has theoretical foundations.  On the negative side, the reviewers were mainly concerned with novelty and also had some concerns about the experiments.  The authors responded to these criticisms (though 3 of 4 reviewers did not respond to the rebuttal; however, I did look at the rebuttal and found it sufficiently convincing as a response to the author concerns).

Overall, the reviewers ultimately all felt fairly positive about the paper and recommended acceptance.

**Justification For Why Not Higher Score:**

There were enough minor concerns about the paper, and none of the reviewers really strongly championed this paper, so I don't think it's strong enough to push to spotlight or oral.

**Justification For Why Not Lower Score:**

All three reviewers were in agreement that the paper is strong enough to be accepted.

---

### Decision · Program_Chairs · 2024-01-16

Accept (poster)